# Learning Input Encodings for Kernel-Optimal Implicit Neural Representations

**Zhemin Li**[1][*]  **Liyuan Ma**[1][*]  **Hongxia Wang**[1]  **Yaoyun Zeng**[1]  **Xiaolong Han**[1]

## Abstract

Implicit Neural Representations (INRs) rely heavily on architectural choices for good generalization. Developing theoretically grounded approaches for architecture design remains an active area of research. Via theoretical analysis of the infinite-width limit, we establish a methodology that characterizes INR's generalization by means of kernel alignment. We first formulate the optimal kernel that minimizes pointwise expected squared error, then demonstrate that the Neural Tangent Kernel of the composed function (INR with input encoding) can approximate any positive semidefinite dot-product kernels through input feature mapping adjustments. Building upon these insights, we propose a Kernel Alignment Regularizer (KAR) that naturally integrates with existing INR systems to enhance kernel alignment. We further develop Plug-in Encoding for Aligned Kernels (PEAK) to refine INR models with KAR using learnable input encoding. This work contributes to the ongoing research efforts in bridging theory and practice for principled INR architecture design. Code is available at https://github.com/lizhemin15/KAR.

## 1. Introduction

Implicit Neural Representation (INR) has emerged as a powerful paradigm for continuous signal modeling (Sitzmann et al., 2020), advancing various domains from computer vision to scientific computing via its ability to handle inverse problems (Mildenhall et al., 2020; Chen et al., 2021; Martin-Brualla et al., 2021; Wadhwani & Kojima, 2022). At its core, INR achieves these successes by leveraging neural networks to map low-dimensional input coordinates directly to output values. Although showing great promise,

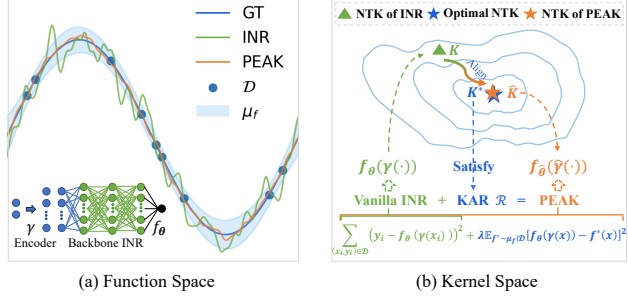

*Figure 1.* Overview of PEAK: (a) Function space representation showing how an INR network $f_\theta$ with a learnable encoder $\gamma$ approximates the Ground Truth, where dark blue dots $\mathcal{D}$ represent training samples and the light blue region represents $\mu_f$, the conditional distribution from which the ground truth function $f^*$ is sampled; (b) Theoretical analysis in kernel space illustrating how the Kernel Alignment Regularizer (KAR) guides the Vanilla INR from the initial kernel $K$ towards the optimal kernel $K^*$. PEAK achieves kernel alignment while maintaining data fidelity through the joint optimization of encoder parameters and INR weights.

the generalization ability of INR can be influenced by architectural choices, including activation functions (Sitzmann et al., 2020; Saragadam et al., 2023), network structures (Fathony et al., 2021; Lindell et al., 2022), and input encodings (Tancik et al., 2020; Müller et al., 2022; Xie et al., 2022; Liu, 2024). Developing systematic strategies for INR architecture design presents an important opportunity for further advancement in the field (Dupont et al., 2022).

We investigate this design problem through theoretical analysis. Due to INR's highly non-linear nature, direct analysis proves difficult. Neural Tangent Kernel (NTK) theory offers a breakthrough by showing that at infinite width, INR's training dynamics can be precisely characterized by kernel regression (Jacot et al., 2018; Chizat et al., 2019). This fundamental connection enables us to leverage well-established kernel method theories to analyze and enhance INR's performance (Tancik et al., 2020; Yüce et al., 2022; Li et al., 2023). Building upon kernel methods, we formulate the optimal kernel that minimizes pointwise expected squared error to measure INR's generalization ability. Finding an INR architecture whose corresponding kernel aligns with the optimal kernel becomes our main focus. This process, known as kernel alignment, has shown promising results in

---

[*]Equal contribution [1]Department of Mathematics, National University of Defense Tecgnology, Changsha, China. Correspondence to: Hongxia Wang <wanghongxia@nudt.edu.cn>.

*Proceedings of the $42^{nd}$ International Conference on Machine Learning*, Vancouver, Canada. PMLR 267, 2025. Copyright 2025 by the author(s).

traditional kernel methods (Bordelon et al., 2020; Cristianini et al., 2001; Jacot et al., 2020) and in aligning kernels between tasks and data (Chapelle et al., 2002; Cortes et al., 2012; Liu et al., 2016; Liu, 2024). Nevertheless, its potential for INR optimization remains to be fully explored.

Matching INR's kernel with the optimal kernel remains a challenging inverse problem. Previous work on NTK reverse engineering has made important theoretical progress, showing that single-layer networks with polynomial activations can approximate any semidefinite dot-product kernel (Simon et al., 2022). However, when moving from theory to practice, deeper architectures have shown particularly promising results (Delalleau & Bengio, 2011; Eldan & Shamir, 2015; Lu et al., 2017), motivating us to explore more flexible architectural choices. In exploring activation functions, designed initialization strategies have proven valuable for maintaining training efficiency and performance (Sitzmann et al., 2020). Drawing from these insights, we focus on input feature mapping since it allows us to leverage existing well-designed deep architectures and their tuned initializations without modification (Tancik et al., 2020; Müller et al., 2022; Xie et al., 2022; Liu, 2024).

Guided by these practical considerations, this work establishes a novel theoretical foundation showing that the Neural Tangent Kernel of INR with input encoding can approximate any positive semidefinite dot-product kernel. Based on this insight, we propose Plug-in Encoding for Aligned Kernels (PEAK), which combines a learnable encoder with a Kernel Alignment Regularizer (KAR) to enhance kernel alignment. PEAK can be readily incorporated into any existing INR system as a plug-in solution without architectural modifications. Figure 1 illustrates this workflow, where the theoretical analysis in kernel space directs the vanilla INR towards the optimal solution using learnable input encoding and kernel alignment.

Our kernel alignment approach shows promising improvements for INR's generalization on inverse problems. The main contributions of this work include:

- **Theoretical Foundation:** A Kernel Alignment Regularizer abbreviated as KAR is proposed in this paper which explores the theoretical connection between INR generalization and kernel alignment, providing a principled approach to enhance INR's performance through kernel alignment.

- **Algorithm Design:** We further propose PEAK algorithm, a plug-in solution that combines learnable encoding with KAR to align INR's kernel with the optimal kernel adaptively.

- **Empirical Validation:** Experiments on image inpainting, phase retrieval, and Neural Radiance Field demonstrate the effectiveness of the proposed approach in improving the generalization of INRs.

## 2. Related Works

**Implicit Neural Representations.** Implicit Neural Representation (INR) offers continuous, differentiable signal encoding, unlike discrete tensor representations that require interpolation and finite differences (Reddy et al., 2021; Sitzmann et al., 2020). This enables efficient physics simulation, shape optimization, and novel view synthesis (Guan et al., 2022; Mildenhall et al., 2020; Martin-Brualla et al., 2021). Research on improving INR's generalization has led to various architectural advances, from activation functions (Xu et al., 2019; Luo et al., 2021; Cao et al., 2021; Vakevičius et al., 2019; Zhao et al., 2019; Sitzmann et al., 2020) to neural network structure (Fathony et al., 2021; Lindell et al., 2022), and input encoding methods (Tancik et al., 2020; Müller et al., 2022; Xie et al., 2022; Liu, 2024). These advances motivate theoretical exploration of architecture design.

**Neural Tangent Kernel Theory.** The NTK framework (Jacot et al., 2018; Chizat et al., 2019) bridges the gap between neural networks and kernel methods by showing that infinitely wide networks behave as kernel regression. This connection enables the theoretical analysis of neural network generalization (Tancik et al., 2020; Li et al., 2023) and expressive power (Yüce et al., 2022). The selection of appropriate kernels for optimal generalization remains a central problem in the field. While single-layer networks with polynomial activations can theoretically approximate any kernel (Simon et al., 2022), recent studies have shown that input feature mapping can significantly alter the NTK's spectral properties (Tancik et al., 2020).

**Kernel Alignment.** Kernel Alignment (KA) has emerged as a powerful principle for improving model generalization by ensuring the kernel used in regression matches the target function's optimal kernel (Bordelon et al., 2020; Cristianini et al., 2001; Jacot et al., 2020). This approach has proven successful in various domains, from kernel parameter tuning (Chapelle et al., 2002) to multiple kernel learning (Cortes et al., 2012) and clustering (Liu et al., 2016; Liu, 2024). Recent theoretical advances have established rigorous connections between kernel alignment and generalization error bounds (Bordelon et al., 2020; Canatar et al., 2021; Wang et al., 2024). Building upon the success of input encoding methods in INR (Tancik et al., 2020; Müller et al., 2022), our work explores the potential synergy between kernel theory and INR practice.

# 3. Plug-in Encoding for Aligned Kernels

Previous works have made significant contributions to INR architecture design using innovative empirical approaches. Drawing from these advances, we aim to complement existing methods by establishing theoretical foundations for achieving optimal generalization via kernel alignment in the infinite-width limit. Stemming from this insight, we propose PEAK to bridge theory and practice using learnable input encoding.

## 3.1. Approximate an Infinite-width INR with Kernel Regression

To establish a theoretical foundation for optimal INR design, we first examine the connection between INRs and kernel methods in the infinite-width limit. Recent theoretical advances have demonstrated that training a neural network under certain conditions resembles a kernel method at infinite width (Jacot et al., 2018; Yang & Littwin, 2021; Golikov et al., 2022). Consider an INR $f_{\boldsymbol{\theta}} : \mathcal{X} \mapsto \mathcal{Y}$ with $L$ hidden layers:

$$
\begin{aligned}
\mathbf{x}^{(\ell)} &= \sigma\left(\mathbf{W}^{(\ell)}\mathbf{x}^{(\ell-1)} + \mathbf{b}^{(\ell)}\right), \quad 1 \le \ell \le L, \\
f_{\boldsymbol{\theta}}(\mathbf{x}) &= \mathbf{W}^{(L+1)}\mathbf{x}^{(L)} + \mathbf{b}^{(L+1)},
\end{aligned} \tag{1}
$$

where $\sigma(\cdot)$ denotes an element-wise activation function, $\mathbf{x}^{(0)} = \mathbf{x}$, $\mathbf{W}^{(\ell)} \in \mathbb{R}^{n_\ell \times n_{\ell-1}}$ and $\mathbf{b}^{(\ell)} \in \mathbb{R}^{n_\ell}$ represent the weight matrix and bias vector of layer $\ell$ respectively, and $\boldsymbol{\theta}$ encompasses all network parameters.

To formalize the learning process, let $\mathcal{X} \subseteq \mathbb{R}^d$ denote the $d$-dimensional input space and $\mathcal{Y} \subseteq \mathbb{R}$ be the output space, i.e., $n_0 = d, n_{L+1} = 1$. Given a training dataset $\mathcal{D} = \{(\boldsymbol{x}_i, y_i) \mid \boldsymbol{x}_i \in \mathcal{X}, y_i \in \mathcal{Y}, i = 1, \dots, N\}$ of size $N$, the INR is trained by minimizing the mean squared error (MSE) loss $\mathcal{L}(\boldsymbol{\theta})$ on $\mathcal{D}$:

$$
\mathcal{L}(\boldsymbol{\theta}) = \frac{1}{2}\sum_{i=1}^{N}(y_i - f_{\boldsymbol{\theta}}(\mathbf{x}_i))^2. \tag{2}
$$

We aim to find the optimal parameters $\boldsymbol{\theta}^*$ such that the trained INR $f_{\boldsymbol{\theta}^*}(\mathbf{x})$ approximates the ground-truth function $f^*(\mathbf{x})$ for any input $\mathbf{x} \in \mathcal{X}$. However, as we all know, it is difficult to find the optimal parameters. Frequently, what we obtain are only sub-optimal solutions (shown in Figure 1), which leads to the generalization problem of INR.

Let us denote the matrix of input samples as $\mathbf{X} = [\mathbf{x}_1, \dots, \mathbf{x}_N]^\top \in \mathbb{R}^{N \times d}$ and the vector of output samples as $\mathbf{Y} = [y_1, \dots, y_N]^\top \in \mathbb{R}^N$. When the width of INR approaches infinity and the training time $t \to \infty$, $f_{\boldsymbol{\theta}_t} \to f_{\boldsymbol{\theta}_\infty}$ which can be characterized by kernel regression (detailed derivation in Appendix A.1):

$$
f_{\boldsymbol{\theta}_\infty}(\mathbf{x}) = K_0(\mathbf{x}, \mathbf{X})K_0^{\dagger}(\mathbf{X}, \mathbf{X})\mathbf{Y}, \tag{3}
$$

where the *Neural Tangent Kernel (NTK)* of this INR is expressed as:

$$
K_0(\mathbf{x}, \mathbf{x}') = \nabla_{\boldsymbol{\theta}_0}^\top f_{\boldsymbol{\theta}_0}(\mathbf{x})\nabla_{\boldsymbol{\theta}_0} f_{\boldsymbol{\theta}_0}(\mathbf{x}'). \tag{4}
$$

Here, $K_0(\mathbf{x}, \mathbf{X})$ represents a row-vector of length $N$ whose $i$-th element is $K_0(\mathbf{x}, \mathbf{x}_i)$ and $K_0^{\dagger}(\mathbf{X}, \mathbf{X}) \in \mathbb{R}^{N \times N}$ whose $i$-th row is $K_0^{\dagger}(\mathbf{x}_i, \mathbf{X})$. $K_0^{\dagger}$ indicates the Moore-Penrose pseudoinverse of matrix $K_0$. The NTK captures the evolution of network predictions during training by measuring how changes in parameters affect the output, providing a powerful tool for analyzing the limiting behavior of INR.

## 3.2. Optimal Kernel

Equation (3) indicates that given an INR $f_{\boldsymbol{\theta}}$, we can characterize its prediction by $f_{\boldsymbol{\theta}_\infty}$ at any $\mathbf{x}$. Next, we examine a pointwise expected squared error as follows,

$$
\mathbb{E}_{f^* \sim \mu_f | \mathcal{D}}\left[(f_{\boldsymbol{\theta}_\infty}(\mathbf{x}) - f^*(\mathbf{x}))^2\right], \tag{5}
$$

where the target function $f^*$ is sampled from a measure $\mu_f$ over an appropriate function space. The key insight of this section lies in formulating the optimal kernel $K^*$ that minimizes this expected error, which will serve as a theoretical guide for designing enhanced INR architectures.

**Theorem 3.1.** *The kernel that minimizes the error defined in equation* (5) *is expressed as*

$$
\mathbf{K}^*(\mathbf{x}, \mathbf{x}') = \mathbb{E}_{f^* \sim \mu_f | \mathcal{D}}\left[f^*(\mathbf{x})f^*(\mathbf{x}')\right]. \tag{6}
$$

*Proof.* To begin with, we introduce a $1 \times N$ vector $\mathbf{M}_{\mathbf{x}} := K_0(\mathbf{x}, \mathbf{X})K_0^{\dagger}(\mathbf{X}, \mathbf{X})$. Substituting equation (3) into the objective function (5) and differentiating with respect to $\mathbf{M}_{\mathbf{x}}$, we obtain

$$
\begin{aligned}
&\nabla_{\mathbf{M}_{\mathbf{x}}}\mathbb{E}_{f^* \sim \mu_f | \mathcal{D}}\left[(f_{\boldsymbol{\theta}_\infty}(\mathbf{x}) - f^*(\mathbf{x}))^2\right] \\
&= \nabla_{\mathbf{M}_{\mathbf{x}}}\mathbb{E}_{f^* \sim \mu_f | \mathcal{D}}\left[(\mathbf{M}_{\mathbf{x}}\mathbf{Y})^2 - 2(\mathbf{M}_{\mathbf{x}}\mathbf{Y})f^*(\mathbf{x}) + (f^*(\mathbf{x}))^2\right] \\
&= 2\mathbf{M}_{\mathbf{x}}\mathbf{Y}\mathbf{Y}^\top - 2\mathbb{E}_{f^* \sim \mu_f | \mathcal{D}}[f^*(\mathbf{x})\mathbf{Y}^\top] = \mathbf{0}.
\end{aligned}
$$

This leads to

$$
\mathbf{M}_{\mathbf{x}}\mathbf{Y}\mathbf{Y}^\top = \mathbb{E}_{f^* \sim \mu_f | \mathcal{D}}[f^*(\mathbf{x})\mathbf{Y}^\top],
$$

and therefore

$$
\mathbf{M}_{\mathbf{x}} = \mathbb{E}_{f^* \sim \mu_f | \mathcal{D}}[f^*(\mathbf{x})\mathbf{Y}^\top](\mathbf{Y}\mathbf{Y}^\top)^{\dagger}. \tag{7}
$$

Note that the Hessian of $\mathbb{E}_{f^* \sim \mu_f | \mathcal{D}}$ with respect to $\mathbf{M}_{\mathbf{x}}$ is

$$
\mathbf{H}_{\mathbf{M}_{\mathbf{x}}} = 2\mathbf{Y}\mathbf{Y}^\top.
$$

As it is positive semi-definite, (5) reaches minimum at $\mathbf{M}_{\mathbf{x}}$ given by (6).

Recalling that $\mathbf{M_x} = K_0(\mathbf{x}, \mathbf{X}) K_0^\dagger(\mathbf{X}, \mathbf{X})$, by comparing with (7) and noting that $\mathbf{Y}\mathbf{Y}^\top = K^*(\mathbf{X}, \mathbf{X})$ and $\mathbb{E}_{f^* \sim \mu_f | \mathcal{D}}[f^*(\mathbf{x})\mathbf{Y}^\top] = K^*(\mathbf{x}, \mathbf{X})$, we arrive at:

$$K_0(\mathbf{x}, \mathbf{X}) K_0^\dagger(\mathbf{X}, \mathbf{X}) = K^*(\mathbf{x}, \mathbf{X}) K^{*\dagger}(\mathbf{X}, \mathbf{X}). \quad (8)$$

This equality holds when

$$K^*(\mathbf{x}, \mathbf{x}') = \mathbb{E}_{f^* \sim \mu_f | \mathcal{D}} \left[ f^*(\mathbf{x}) f^*(\mathbf{x}') \right], \quad (9)$$

which completes the proof. □

*Remark* 3.2. In this context, we employ the Moore-Penrose pseudo-inverse rather than the regular inverse because the kernel matrices may be singular or ill-conditioned in practice. Additionally, $K^*$ in equation (6) is not unique, as the scaled version of $K^*$, namely $\alpha K^*, \alpha > 0$, yields the same $\mathbf{M_x} = K^*(\mathbf{x}, \mathbf{X}) K^{*\dagger}(\mathbf{X}, \mathbf{X})$.

Since $f^*$ is sampled from $\mu_f$ conditioned on $\mathcal{D}$, for any $\mathbf{x} \in \mathcal{X}$ and training sample $\mathbf{x}_i \in \mathcal{S} = \{\boldsymbol{x}_i\}_{i=1}^N$, where we deliberately introduce $\mathcal{S}$ to distinguish between the entire domain $\mathcal{X}$ and the training set,

$$K^*(\mathbf{x}, \mathbf{x}_i) = \mathbb{E}_{f^* \sim \mu_f | \mathcal{D}} \left[ f^*(\mathbf{x}) f^*(\mathbf{x}_i) \right]$$
$$= y_i \mathbb{E}_{f^* \sim \mu_f | \mathcal{D}} [f^*(\mathbf{x})].$$

Specifically, for $\mathbf{x}_i, \mathbf{x}_j \in \mathcal{S}$, we obtain

$$K^*(\mathbf{x}_j, \mathbf{x}_i) = \mathbb{E}_{f^* \sim \mu_f | \mathcal{D}} \left[ f^*(\mathbf{x}_j) f^*(\mathbf{x}_i) \right] = y_i y_j.$$

Utilizing kernel regression with the optimal kernel $K^*$ formulated in Theorem 3.1 faces two fundamental limitations. Computing $K^*(\mathbf{x}, \mathbf{x}') = \mathbb{E}_{f^* \sim \mu_f | \mathcal{D}} \left[ f^*(\mathbf{x}) f^*(\mathbf{x}') \right]$ remains intractable in practice due to lack of direct access to $\mu_f$. Moreover, even with $K^*$ available, evaluating $K^{*\dagger}$ requires $\mathcal{O}(N^3)$ operations, posing computational constraints for large-scale datasets. In comparison, INRs provide an alternative approach that can potentially handle various inverse problems with $\mathcal{O}(N)$ complexity during forward propagation. Therefore, we propose to improve INR performance by incorporating theoretical insights from optimal kernel selection while circumventing the computational overhead of explicit kernel regression.

### 3.3. Optimal INR

Without loss of generality, we assume each input $\mathbf{x}_i$ satisfies $\|\mathbf{x}_i\|_2 = 1$. This condition can be achieved utilizing a feature map $F(\mathbf{x}) = [\cos \mathbf{x}, \sin \mathbf{x}]$ which ensures $\|F(\mathbf{x}_i)\|_2 = 1$. Under this normalization, the NTK exhibits rotation-invariance, meaning $K(\mathbf{x}, \mathbf{x}') = K(c)$ with $c = \mathbf{x}^\top \mathbf{x}' \in [-1, 1]$. For notational simplicity, we employ the same $K$ to denote both the functions of one variable and two variables. We refer to kernels of this form as *dot-product*

*kernels* (Jacot et al., 2018). Reverse engineering shows that any dot-product kernel can be approximated by carefully initializing a single-hidden-layer INR (Simon et al., 2022). Specifically, it requires specialized initialization schemes and restricts us to shallow architectures, preventing us from leveraging the benefits of modern deep networks.

A more practical solution involves utilizing a feature map encoder $\gamma(\mathbf{x}) : \mathcal{X} \to \mathbb{R}^{d_e}$ (Tancik et al., 2020; Müller et al., 2022; Liu, 2024). In contrast to the reverse engineering approach, this encoder can be seamlessly integrated into any existing deep architecture without modifying their initialization or depth. The resulting INR becomes $f_{\boldsymbol{\theta}}(\gamma(\mathbf{x}))$, with its NTK $K_\gamma(\mathbf{x}, \mathbf{x}') = K(\gamma(\mathbf{x}), \gamma(\mathbf{x}'))$. The following theorem demonstrates that with appropriate design, such a learnable encoder can achieve the optimal kernel $K^*$:

**Theorem 3.3.** *Given dot-product kernels $K, K^*$ (see Lemma A.1 for detailed properties), where $K : [-1, 1] \to B \subseteq \mathbb{R}$ represents a Lipschitz continuous bijection with Lipschitz constant $C_K$ and $K^*$ is continuous on $[-1, 1]$ (i.e., $K^* \in C([-1,1])$) satisfying $K^*[-1, 1] \subseteq B^* \subseteq \mathbb{R}$. If $B^* \subseteq B$, then for any $\varepsilon > 0$, there exists a mapping $\gamma_\varepsilon : \mathcal{X} \to \mathbb{R}^{\sum_{k=0}^{N_\varepsilon} d^k}$ such that*

$$\sup_{c \in [-1,1]} |K_{\gamma_\varepsilon}(c) - K^*(c)| < \varepsilon,$$

*where $K_{\gamma_\varepsilon}(c) = K(\gamma_\varepsilon(\mathbf{x}), \gamma_\varepsilon(\mathbf{x}'))$ with $c = \mathbf{x}^\top \mathbf{x}'$ for any unit vectors $\mathbf{x}, \mathbf{x}' \in \mathcal{X}$.*

*Proof.* Since $K$ represents a bijection, its inverse $K^{-1} : B \to [-1, 1]$ exists and $h(c) = K^{-1}(K^*(c))$ maps $[-1, 1]$ to itself with $K(h(c)) = K^*(c)$. As the composition of dot-product kernels, $h(c)$ is clearly a dot-product kernel. By Theorem A.2, $h(c)$ admits a non-negative power series:

$$h(c) = \sum_{k=0}^{\infty} a_k c^k, \quad a_k \geq 0, \ \sum_{k=0}^{\infty} a_k < \infty.$$

For any $\varepsilon > 0$, choose $N_\varepsilon$ such that $\sum_{k > N_\varepsilon} a_k < \varepsilon / C_K$ and construct:

$$\gamma_\varepsilon(\mathbf{x}) = \bigoplus_{k=0}^{N_\varepsilon} \sqrt{a_k} \mathbf{x}^{\otimes k} \in \mathbb{R}^{\sum_{k=0}^{N_\varepsilon} d^k},$$

where $\mathbf{x}^{\otimes k}$ denotes the $k$-fold tensor product, $\oplus$ and $\otimes$ represent the direct sum and direct product, respectively. For unit vectors $\mathbf{x}, \mathbf{x}' \in \mathcal{X}$:

$$\gamma_\varepsilon(\mathbf{x})^\top \gamma_\varepsilon(\mathbf{x}') = \sum_{k=0}^{N_\varepsilon} a_k (\mathbf{x}^\top \mathbf{x}')^k.$$

**Algorithm 1** PEAK Training Algorithm

---

**Input:** INR $f_{\boldsymbol{\theta}}$, attention network $g_{\boldsymbol{\theta}'}$, polynomial encoder $\gamma$, initial parameters $\{\boldsymbol{\theta}_0, \boldsymbol{\theta}'_0, \{a_j(0)\}_{j=0}^{N_\gamma}\}$, training data $\mathcal{D}$, grid points $\mathbf{X}_G$, loss $\mathcal{L}_{\text{all}}$, learning rate $\alpha$, epochs $T$

**for** $t = 0 : T - 1$ **do**

    $\mathcal{L}_{\text{all}}(t) \leftarrow \mathcal{L}_{\text{all}}(\boldsymbol{\theta}_t, \{a_j(t)\}_{j=0}^{N_\gamma}, \boldsymbol{\theta}'_t)$ {Eq. (12)}

    $\boldsymbol{\theta}_{t+1} \leftarrow \boldsymbol{\theta}_t - \alpha \nabla_{\boldsymbol{\theta}_t} \mathcal{L}_{\text{all}}(t)$

    $\boldsymbol{\theta}'_{t+1} \leftarrow \boldsymbol{\theta}'_t - \alpha \nabla_{\boldsymbol{\theta}'_t} \mathcal{L}_{\text{all}}(t)$

    **for** $j = 0 : N_\gamma$ **do**

        $a_j(t + 1) \leftarrow a_j(t) - \alpha \nabla_{a_j(t)} \mathcal{L}_{\text{all}}(t)$

    **end for**

**end for**

**return** $f_{\boldsymbol{\theta}_T}(\gamma(\mathbf{x}, \{a_j(T)\}_{j=0}^{N_\gamma})), g_{\boldsymbol{\theta}'_T}$

---

By Lipschitz continuity of $K$:

$$\left| K_{\gamma_\varepsilon}(c) - K^*(c) \right| = \left| K\left( \sum_{k=0}^{N_\varepsilon} a_k c^k \right) - K\left( \sum_{k=0}^{\infty} a_k c^k \right) \right|$$
$$\leq C_K \sum_{k > N_\varepsilon} a_k < \varepsilon,$$

where the first inequality follows from $|c| \leq 1$, $a_k \geq 0$. $\quad \square$

*Remark* 3.4. The key insight enabling this construction is the tensor product property: $(\mathbf{x}_1^{\otimes k})^\top (\mathbf{x}_2^{\otimes k}) = (\mathbf{x}_1^\top \mathbf{x}_2)^k$ for unit vectors. When $N_\varepsilon = 1$, the encoder takes a simple form $\gamma_\varepsilon(\mathbf{x}) = [\sqrt{a_0}, \sqrt{a_1}\mathbf{x}]^\top$. This often suffices in practice since higher-order terms ($k > 2$) contribute less as $c^k = \cos^k \theta$ rapidly approaches zero, particularly in scenarios where $\mathbf{x}_1^\top \mathbf{x}_2 < 1$.

*Remark* 3.5. The construction is not unique. We can decompose $\sqrt{a_0}$ into $[\sqrt{a_0 \alpha_1}, \ldots, \sqrt{a_0 \alpha_m}]$ for any partition $\{\alpha_i\}_{i=1}^m$ with $\sum_{i=1}^m \alpha_i = 1$, which yields: $\gamma_\varepsilon(\mathbf{x}) = [\sqrt{a_0 \alpha_1}, \ldots, \sqrt{a_0 \alpha_m}] \oplus \bigoplus_{k=1}^{N_\varepsilon} \sqrt{a_k} \mathbf{x}^{\otimes k}$.

### 3.4. The PEAK Algorithm

Although Theorem 3.3 offers theoretical guidance, direct computation remains challenging since we cannot directly access the underlying distribution $\mu_f$. Nevertheless, we can leverage a key insight: $f^*(\mathbf{x})$ can be characterized via its relationships with $\mathbf{Y}$. Following this intuition, we model $\mu_f$ as a discrete distribution where $\mathbb{P}(f^*(\mathbf{x}) = y_i) = \mathbf{A}(\mathbf{x}, \mathbf{x}_i)$ for $i = 1, 2, \ldots, N$, with $\sum_{i=1}^N \mathbf{A}(\mathbf{x}, \mathbf{x}_i) = 1$ and $\mathbf{A}(\mathbf{x}, \mathbf{x}') \geq 0$ ($\forall \mathbf{x}, \mathbf{x}' \in \mathcal{X}$) denotes a function that measures the similarity between $\mathbf{x}$ and $\mathbf{x}'$. Combining equations (3) and (6),

we obtain

$$\begin{aligned} f_{\boldsymbol{\theta}_\infty}(\mathbf{x}) &= K^*(\mathbf{x}, \mathbf{X}) K^{*\dagger}(\mathbf{X}, \mathbf{X}) \mathbf{Y} \\ &= \mathbb{E}_{f^* \sim \mu_f | \mathcal{D}}[f^*(\mathbf{x})] \cdot \mathbf{Y}^\top (\mathbf{Y}\mathbf{Y}^\top)^\dagger \mathbf{Y} \\ &= \mathbb{E}_{f^* \sim \mu_f | \mathcal{D}}[f^*(\mathbf{x})] = \sum_{i=1}^N \mathbf{A}(\mathbf{x}, \mathbf{x}_i) y_i. \end{aligned} \quad (10)$$

It is worth noting that $\mathbb{E}_{f^* \sim \mu_f | \mathcal{D}}[f^*(\mathbf{x})]$ resides in the row space of $\mathbf{Y}$. According to the properties of the Moore-Penrose pseudoinverse, for any vector $\mathbf{v}$ within the row space of $\mathbf{Y}$, we have $\mathbf{v}\mathbf{Y}^\top (\mathbf{Y}\mathbf{Y}^\top)^\dagger \mathbf{Y} = \mathbf{v}$. This elucidates why the second-to-last equality in equation (10) holds.

In numerous practical applications, such as NeRF (Mildenhall et al., 2020), the ground truth $y_i$ is not directly observable. In these scenarios, $f_{\boldsymbol{\theta}_\infty}$ also remains unknown since it depends on $y_i$. However, we recognize that $f_{\boldsymbol{\theta}_t}(\mathbf{x}_i)$ should converge to the true $y_i$ as $t \to \infty$. Consequently, we can substitute both $f_{\boldsymbol{\theta}_\infty}$ and $y_i$ in equation (10) with $f_{\boldsymbol{\theta}_t}$ (denoted as $f_{\boldsymbol{\theta}}$ for simplicity), yielding $f_{\boldsymbol{\theta}}(\mathbf{x}) = \sum_{i=1}^N \mathbf{A}(\mathbf{x}, \mathbf{x}_i) f_{\boldsymbol{\theta}}(\mathbf{x}_i)$. When incorporating the encoder $\gamma$, this relationship suggests that the optimal INR $f_{\boldsymbol{\theta}}$ should satisfy

$$\mathcal{R}(f_{\boldsymbol{\theta}}(\gamma(\mathbf{x})), \mathbf{A}) = \|f_{\boldsymbol{\theta}}(\gamma(\mathbf{x})) - \mathbf{A}(\mathbf{x}, \mathbf{X}) f_{\boldsymbol{\theta}}(\gamma(\mathbf{X}))\|_p = 0,$$

for some vector norm $\|\cdot\|_p$, where $\mathbf{A}(\mathbf{x}, \mathbf{X}) \in \mathbb{R}^{1 \times N}$. We designate this term as the ***Kernel Alignment Regularizer (KAR)***, as it enforces alignment between the INR's kernel and the optimal kernel derived from our theoretical analysis. In this work, we adopt $\|\cdot\|_2$ as our choice of norm.

Consequently, to achieve optimal INR representation (in terms of kernel alignment) while maintaining fidelity on training data, we formulate the following optimization problem:

$$\min_{\boldsymbol{\theta}, \gamma, \mathbf{A}} \sum_{i=1}^N \mathcal{L}(f_{\boldsymbol{\theta}}(\gamma(\mathbf{x}_i)), y_i) + \lambda \int_{\mathcal{X}} \mathcal{R}(f_{\boldsymbol{\theta}}(\gamma(\mathbf{x})), \mathbf{A}) d\mathbf{x}, \quad (11)$$

where $\gamma$ serves as a learnable encoder for kernel alignment and $\lambda$ represents a trade-off hyperparameter. The first term ensures fidelity on $\mathcal{S}$, and the integral term constitutes the continuous form of KAR that enforces kernel alignment across the entire input space $\mathcal{X}$.

We observe that both $\gamma$ and $\mathbf{A}$ possess inherent structural constraints that must be satisfied, necessitating their parameterization in a structure-preserving manner. Additionally, the integral in equation (11) requires discretization for practical implementation. Following the construction established in the proof of Theorem 3.3, we express: $\gamma(\mathbf{x}) = \bigoplus_{k=0}^{N_\gamma} \sqrt{a_k} \mathbf{x}^{\otimes k}$, where $\mathbf{x}^{\otimes k}$ signifies the $k$-fold tensor product of $\mathbf{x}$ (when $k = 0$, it reduces to the scalar $\sqrt{a_0}$). In practical applications, we discover that setting $N_\gamma = 1$ yields satisfactory performance in realistic image

tasks, resulting in $\gamma(\mathbf{x}) = [\sqrt{a_0}, \sqrt{a_1}\mathbf{x}]^\top$. This simplification proves reasonable since higher-order terms contribute diminishingly to the kernel due to $(\mathbf{x}_1^\top\mathbf{x}_2)^k$ rapidly approaching zero for larger $k$.

The function $\mathbf{A}(\mathbf{x}, \mathbf{x}')$ should exhibit continuity in mapping from $\mathcal{X} \times \mathcal{X}$ to $\mathbb{R}$, while satisfying two key conditions: (a) $\sum_{i=1}^{N} \mathbf{A}(\mathbf{x}, \mathbf{x}_i) = 1$, and (b) $\mathbf{A}(\mathbf{x}, \mathbf{x}') \geq 0, \forall \mathbf{x}, \mathbf{x}' \in \mathcal{X}$. We parameterize $\mathbf{A}$ as:

$$\mathbf{A}_{\boldsymbol{\theta}'}(\mathbf{x}, \mathbf{X}) = \mathrm{softmax}(g_{\boldsymbol{\theta}'}(\mathbf{x})^\top g_{\boldsymbol{\theta}'}(\mathbf{X})),$$

where $g_{\boldsymbol{\theta}'} : \mathbb{R}^d \to \mathbb{R}^r$ is implemented as a compact INR. This elegant formulation naturally satisfies both conditions while enabling flexible coordinate relationships.

Taking 2D image processing as an illustrative example, we can sample a uniform discrete grid $\mathbf{X}_G$ from $\mathcal{X}$ with dimensions $m \times n$, where $\mathbf{X}_G = \left\{\left(\frac{i}{m}, \frac{j}{n}\right) \mid 1 \leq i \leq m, 1 \leq j \leq n\right\} \subseteq \mathcal{X}$. The complete loss function can be reformulated as:

$$\mathcal{L}_{\mathrm{all}} = \min_{\boldsymbol{\theta}, \{a_j\}_{j=0}^{N_\gamma}, \boldsymbol{\theta}'} \sum_{i=1}^{N} \mathcal{L}(f_{\boldsymbol{\theta}}(\gamma(\mathbf{x}_i, \{a_j\}_{j=0}^{N_\gamma})), y_i)$$
$$+ \lambda \sum_{\mathbf{x} \in \mathbf{X}_G} \mathcal{R}(f_{\boldsymbol{\theta}}(\gamma(\mathbf{x}, \{a_j\}_{j=0}^{N_\gamma})), \mathbf{A}_{\boldsymbol{\theta}'}),$$

(12)

where the second term represents the discretized KAR that ensures kernel alignment on the uniform grid $\mathbf{X}_G$.

The optimization of equation (12) becomes feasible since it exhibits differentiability with respect to parameters $\boldsymbol{\theta}, \{a_j\}_{j=0}^{N_\gamma}, \boldsymbol{\theta}'$ when $\mathcal{L}, f_{\boldsymbol{\theta}}$ and $g_{\boldsymbol{\theta}}'$ are differentiable with respect to these parameters. This differentiability property holds for common loss functions such as $L_2$ and INRs equipped with differentiable activation functions like $\sin(\cdot)$. Consequently, we can employ gradient-based optimization techniques, such as gradient descent, to minimize equation (12). Upon convergence, we obtain the optimal parameters $\boldsymbol{\theta}^*, \{a_j^*\}_{j=0}^{N_\gamma}, \boldsymbol{\theta}'^*$, yielding the optimized INR $f_{\boldsymbol{\theta}^*}(\gamma(\mathbf{x}, \{a_j^*\}_{j=0}^{N_\gamma}))$. Given that $\mathcal{R}$ facilitates kernel alignment by guiding the encoder $\gamma$ towards the kernel regression solution, we designate this methodology as the **_Plug-in Encoding for Aligned Kernels (PEAK)_** algorithm. For notational clarity, we present the complete algorithm in Algorithm 1.

The optimal INR is encompassed within equation (11), yet not all solutions to equation (11) necessarily represent the optimal INR. Thus, equation (11) serves as a necessary condition for the optimal INR rather than a sufficient one. Notably, our experimental results (Figure 2) demonstrate that the INRs discovered through our algorithm closely approximate the optimal kernel.

**Proposition 3.6.** *Let $\mathbf{X}_G = \mathbf{X}$ and define $\mathcal{R}(f_{\boldsymbol{\theta}}(\gamma(\mathbf{x})), \mathbf{A})$*

*as*

$$\mathcal{R}(f_{\boldsymbol{\theta}}(\gamma(\mathbf{x})), \mathbf{A}) = \|f_{\boldsymbol{\theta}}(\gamma(\mathbf{x})) - \mathbf{A}(\mathbf{x}, \mathbf{X})f_{\boldsymbol{\theta}}(\gamma(\mathbf{X}))\|_2 .$$

*Then, the following equality holds:*

$$\sum_{\mathbf{x} \in \mathbf{X}_G} \mathcal{R}(f_{\boldsymbol{\theta}}(\gamma(\mathbf{x})), \mathbf{A}) = \|(\mathbf{I} - \mathbf{A}(\mathbf{X}, \mathbf{X}))f_{\boldsymbol{\theta}}(\gamma(\mathbf{X}))\|_2 .$$

*This result reveals that KAR corresponds to the Dirichlet Energy (Li et al., 2022). From a geometric perspective, the learned function $\mathbf{A}(\mathbf{x}, \mathbf{x}')$ characterizes the similarity between points $\mathbf{x}$ and $\mathbf{x}'$. Our framework provides a more comprehensive formulation of this relationship through the lens of kernel alignment.*

## 4. Experiments

We evaluate PEAK from three aspects: (1) verifying the kernel alignment properties of PEAK, (2) analyzing the impact of architectural choices, and (3) examining performance in both linear and nonlinear inverse problems, comparing with strong baseline methods, including vanilla MLP (ReLU activation), Fourier feature networks (Fourier) (Tancik et al., 2020), and Hash (Müller et al., 2022).

### 4.1. Kernel Alignment Verification

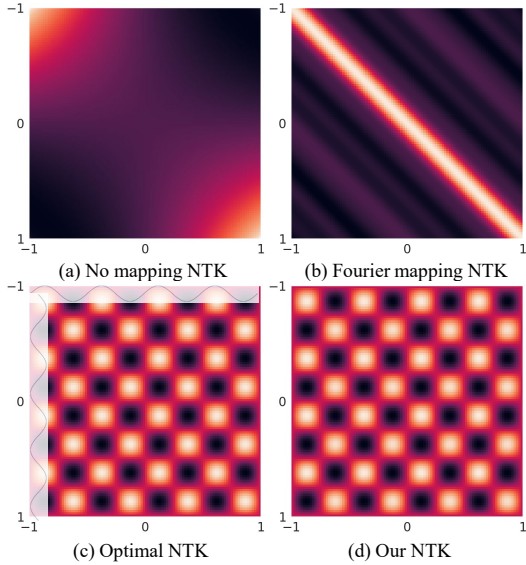

(a) No mapping NTK     (b) Fourier mapping NTK

(c) Optimal NTK     (d) Our NTK

*Figure 2.* Visualization of Neural Tangent Kernels (NTK) for different architectures on the interval $[-1, 1] \times [-1, 1]$.

To examine how PEAK influences kernel alignment, we conduct experiments on a 1D signal fitting task where we can numerically compute both kernels. We consider a synthetic signal $f^*(\mathbf{x}) = \sin(4\pi\mathbf{x})$ with $\mathbf{x} \in [-1, 1]$. The optimal kernel $K^*$ is computed using equation (6), while INR's empirical kernel is calculated as $K(\mathbf{x}, \mathbf{x}') = \nabla_{\boldsymbol{\theta}}^\top f_{\boldsymbol{\theta}}(\mathbf{x})\nabla_{\boldsymbol{\theta}} f_{\boldsymbol{\theta}}(\mathbf{x}')$.

As shown in Figure 2, standard INR with Fourier features (Figure 2(b)) exhibits shift-invariant properties that are beneficial for many tasks. PEAK learns a kernel (Figure 2(d)) that shares characteristics with the optimal kernel (Figure 2(c)), suggesting the effectiveness of our kernel alignment approach.

### 4.2. Impact of Polynomial Degree

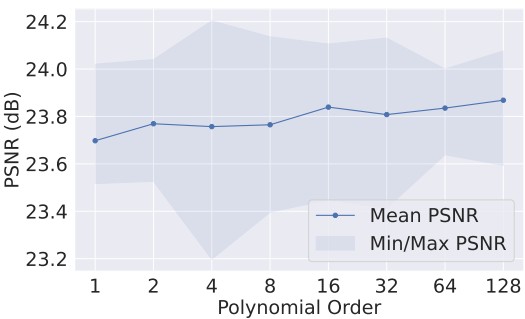

*Figure 3.* Analysis of polynomial order's impact on reconstruction quality. The blue curve shows mean PSNR values, with shaded regions indicating min/max variations.

Following Theorem 3.3, we study how the polynomial degree $N_\gamma$ in encoder $\gamma$ affects approximation accuracy. We conduct experiments on the Jetplane image using the image reconstruction task with combined random and structured missing patterns. We evaluate polynomial degrees from $N_\gamma = 1$ to $N_\gamma = 128$. Results in Figure 3 show that while higher degrees can improve performance, the gains become modest after $N_\gamma = 1$, suggesting that balancing performance and complexity is a practical choice. Furthermore, the influences of the regularization coefficient $\lambda$, the output dimension $r$ of the regularization network, and the activation function in PEAK are thoroughly examined in Appendix B.1.

### 4.3. Linear Inverse Problem: Image Reconstruction

We evaluate PEAK on image reconstruction with missing data. This involves predicting unseen regions from partial observations. The linearity of this problem provides a clear testbed for studying INR's generalization in the basic setting. We test three complex scenarios with missing patterns shown in Figure 4(a): Random (50% pixels randomly removed), Patch (structured regions missing), and Textural (complex patterns missing).

Results in Table 1 indicate that PEAK improves across different scenarios. For the Baboon image with random missing pixels, PEAK achieves 22.60dB PSNR compared to 18.64dB (MLP), 20.10dB (Fourier), and 19.67dB (Hash). Similar trends are observed for structured missing patterns, where PEAK reaches 35.20dB PSNR on the Cameraman

*Table 1.* PSNR (dB) of reconstructed images by INRs. Results from six standard test images under three missing types: random, patch, and textural.

| Image | Missing Type | MLP | Fourier | Hash | PEAK |
|---|---|---|---|---|---|
| Baboon | Random | 18.64 | 20.10 | 19.67 | **22.60** |
| | Patch | 18.82 | 22.35 | 25.44 | **36.08** |
| | Textural | 18.69 | 23.77 | 28.84 | **31.41** |
| Boat | Random | 22.38 | 25.17 | 26.57 | **26.85** |
| | Patch | 22.22 | 27.70 | 26.52 | **33.26** |
| | Textural | 22.79 | 29.68 | 33.85 | **38.96** |
| Cameraman | Random | 23.87 | 26.17 | 28.51 | **29.06** |
| | Patch | 23.71 | 28.68 | 28.19 | **35.20** |
| | Textural | 24.03 | 30.03 | 33.07 | **38.02** |
| Jetplane | Random | 21.59 | 25.39 | 25.26 | **27.55** |
| | Patch | 21.61 | 26.09 | 26.59 | **34.40** |
| | Textural | 21.10 | 30.67 | 32.65 | **36.94** |
| Lake | Random | 20.01 | 23.52 | 24.40 | **26.65** |
| | Patch | 19.71 | 24.41 | 21.58 | **31.93** |
| | Textural | 19.66 | 26.53 | 32.78 | **34.08** |
| Livingroom | Random | 22.40 | 25.52 | 27.32 | **27.95** |
| | Patch | 21.99 | 24.21 | 27.63 | **33.76** |
| | Textural | 22.03 | 30.00 | 34.20 | **38.15** |

image with patch missing, compared to 23.71dB (MLP), 28.68dB (Fourier), and 28.19dB (Hash). Visual results in Figure 4 also show that PEAK helps maintain fine textures and natural transitions in the reconstructed regions.

### 4.4. Nonlinear Inverse Problem: Phase Retrieval

*Table 2.* PSNR (dB) achieved by various neural network based phase retrieval methods for FPR and GPR with different sample ratios $s$.

| | Image | $s$ | MLP | Fourier | Hash | PEAK |
|---|---|---|---|---|---|---|
| FPR (Net-ADM) | House | 1.9 | 28.02 | 16.25 | 31.75 | **40.44** |
| | | 1.8 | 28.72 | 17.40 | 28.78 | **31.77** |
| | | 1.7 | 26.76 | 16.92 | 25.38 | **28.11** |
| | Boston | 1.9 | 17.73 | 16.64 | 18.88 | **21.82** |
| | | 1.8 | 17.37 | 15.21 | 17.86 | **18.75** |
| | | 1.7 | 18.13 | 13.08 | 15.32 | **18.83** |
| | Boat | 1.9 | 22.83 | 20.90 | 28.96 | **33.47** |
| | | 1.8 | 22.00 | 15.22 | 23.79 | **30.52** |
| | | 1.7 | 22.12 | 13.61 | 22.22 | **24.90** |
| GPR (Net-GD) | House | 1.0 | 19.14 | 21.88 | 23.03 | **33.19** |
| | | 0.9 | 19.03 | 21.57 | 20.81 | **33.25** |
| | | 0.8 | 19.05 | 20.98 | 19.20 | **32.65** |
| | Boston | 1.0 | 17.95 | 19.27 | 17.56 | **24.97** |
| | | 0.9 | 17.97 | 19.03 | 16.08 | **24.74** |
| | | 0.8 | 17.93 | 18.59 | 15.63 | **22.25** |
| | Boat | 1.0 | 19.97 | 21.18 | 19.00 | **29.92** |
| | | 0.9 | 19.91 | 20.85 | 17.05 | **30.05** |
| | | 0.8 | 19.95 | 20.36 | 15.95 | **29.10** |

We evaluate PEAK on the nonlinear inverse problem of phase retrieval (PR), which reconstructs a signal using only magnitude measurements obtained from the Fourier transform, referred to as Fourier phase retrieval (FPR), or Gaus-

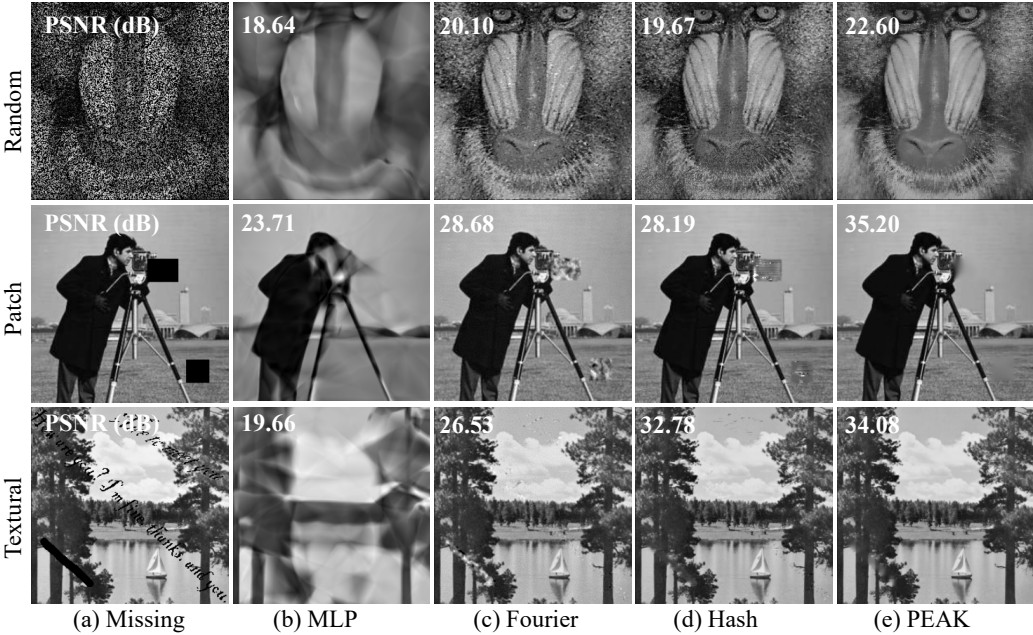

Figure 4. Qualitative comparison of image reconstruction results. Each row represents a different scenario: Baboon with random 50% missing pixels (top), Cameraman with large patch missing (middle), and Lake with textural pattern missing (bottom).

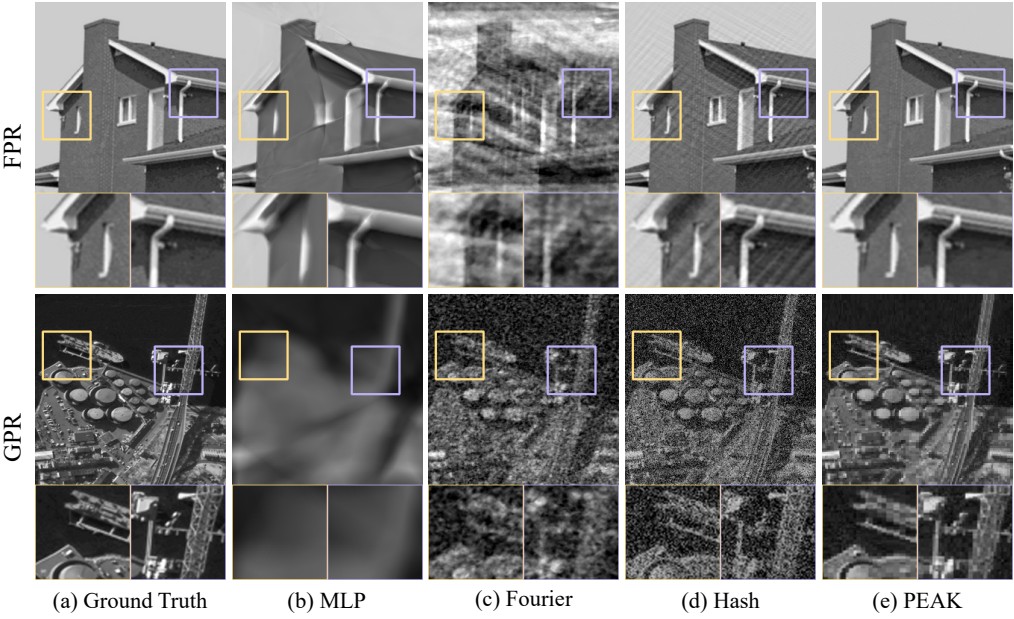

Figure 5. The first row displays the reconstruction images (House) of FPR by Net-ADM at 1.9 sampling ratio, while the second row displays the reconstructions (Boston) of GPR by Net-GD at 1.0.

sian matrix linear operations, known as Gaussian phase retrieval (GPR). A reduced number of measurements increases the challenge of PR, with the sampling ratio $s$ defined as the ratio of measurements to the signal length in each dimension. Building on this foundation, Net-GD (Jagatap & Hegde, 2019a;b) and Net-ADM (Ma et al., 2024) com-

bine a deep decoder with GD and ADMM, respectively (see Appendix B.2). Since both the deep decoder and INR are untrained networks, we replace the former with an INR in Net-GD and Net-ADM, testing different input encodings.

Results in Table 2 show that PEAK provides improvements

of up to 8.69dB in FPR and 11.68dB in GPR, while maintaining stable performance at lower sampling rates. Visual results in Figure 5 indicate that PEAK helps reconstruct both global image structure and local details. These results suggest the potential benefits of kernel alignment in nonlinear inverse problems.

### 4.5. Neural Radiance Field

*Table 3.* Average PSNR (dB) of NeRF by Instant-NGP and our proposed PEAK under different numbers of view perspective samples.

| Synthetic view | Sample numbers | Instant-NGP | PEAK |
|---|---|---|---|
| chair | 100 | **30.60** | 30.32 |
| | 50 | 24.97 | **28.34** |
| | 25 | 23.79 | **24.42** |
| hotdog | 100 | 32.50 | **32.76** |
| | 50 | 29.85 | **31.34** |
| | 25 | 23.32 | **24.51** |
| lego | 100 | 27.98 | **28.01** |
| | 50 | 26.23 | **27.03** |
| | 25 | 21.74 | **23.27** |
| drums | 100 | 23.93 | **24.19** |
| | 50 | 19.43 | **20.08** |
| | 25 | 16.92 | **17.66** |
| ficus | 100 | 24.07 | **24.08** |
| | 50 | 21.48 | **21.86** |
| | 25 | 18.75 | **19.65** |
| mic | 100 | 28.35 | **29.94** |
| | 50 | 25.17 | **26.87** |
| | 25 | 21.72 | **22.39** |

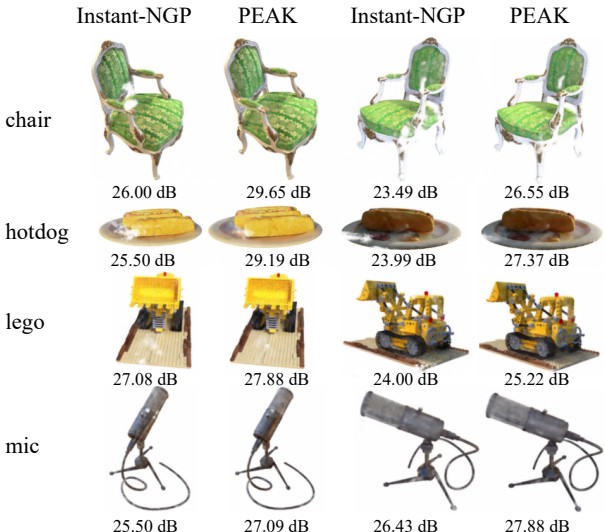

*Figure 6.* Visual results of Neural Radiance Field by Instant-NGP and our proposed PEAK with 50 view perspectives.

Neural Radiance Field (NeRF) synthesizes novel views from multi-view images. We compare PEAK with Instant-NGP

(Müller et al., 2022) on NeRF using 25, 50, and 100 input views from the NeRF synthetic dataset. PEAK outperforms vanilla Instant-NGP in PSNR (25 views, Table 3), especially with fewer samples. Figure 6 shows PEAK reduces sparse-sample artifacts, demonstrating better generalization.

### 4.6. The Computational Efficiency of PEAK

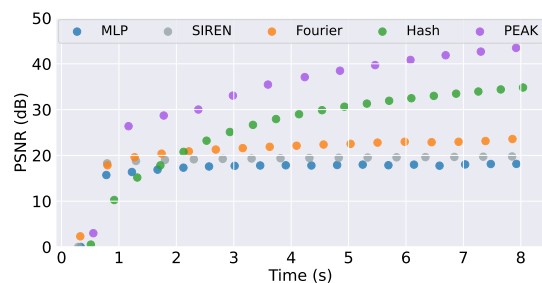

*Figure 7.* PSNR (dB) of INRs for the Baboon image reconstruction over time.

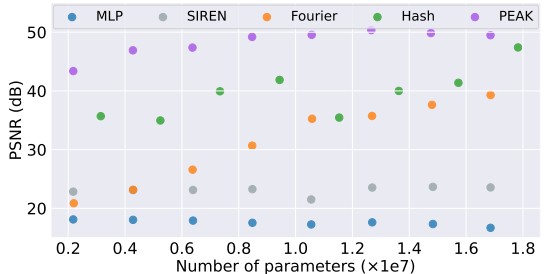

*Figure 8.* PSNR (dB) of INRs for the Baboon image reconstruction with varying number of parameters.

We compare PEAK's efficiency with MLP, SIREN (Sitzmann et al., 2020), Fourier, and Hash on Baboon image reconstruction. PEAK trains faster (Figure 7) and achieves higher PSNR with fewer parameters (Figure 8), outperforming alternatives in speed and performance.

## 5. Concluding Remarks

In this work, we have presented PEAK, a principled framework for enhancing INR's generalization through kernel-guided design. Looking forward, we believe understanding the theoretical relationship between kernel alignment and INR's expressiveness could provide deeper insights into neural network design.

## Impact Statement

This paper presents work whose goal is to advance the field of Machine Learning. There are many potential societal consequences of our work, none which we feel must be specifically highlighted here.

## Acknowledgements

We sincerely thank the anonymous reviewers for their valuable and constructive feedback, which has greatly improved our paper. This work was supported by the following grants: Project 2020YFA0713504 under the National Key Research and Development Program of China, and Grant 12471401 from the National Natural Science Foundation of China.

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

# Appendix

## A. Theoretical Analysis

### A.1. Kernel Regression and Neural Tangent Kernel

Let $\mathcal{X} \subseteq \mathbb{R}^d$ denote a $d$-dimensional input space and $\mathcal{Y} \subseteq \mathbb{R}$ be the output space. For a dataset $\mathcal{D} = \{(\boldsymbol{x}_i, y_i) \mid \boldsymbol{x}_i \in \mathcal{X}, y_i \in \mathcal{Y}, i = 1, \ldots, N\}$ of size $N$, we denote $\mathbf{X} = [\mathbf{x}_1, \ldots, \mathbf{x}_N]^\top \in \mathbb{R}^{N \times d}$ and $\mathbf{Y} = [y_1, \ldots, y_N]^\top \in \mathbb{R}^N$. The ground-truth function is denoted as $f^* : \mathcal{X} \to \mathcal{Y}$.

#### A.1.1. KERNEL REGRESSION FRAMEWORK

We start from the mean squared error (MSE) loss function:

$$\mathcal{L}(\boldsymbol{\theta}) = \frac{1}{2} \sum_{i=1}^{N} (y_i - f_{\boldsymbol{\theta}}(\mathbf{x}_i))^2. \tag{13}$$

The gradient of the loss with respect to the parameters $\boldsymbol{\theta}$ is:

$$\nabla_{\boldsymbol{\theta}} \mathcal{L}(\boldsymbol{\theta}) = -\sum_{i=1}^{N} (y_i - f_{\boldsymbol{\theta}}(\mathbf{x}_i)) \nabla_{\boldsymbol{\theta}} f_{\boldsymbol{\theta}}(\mathbf{x}_i). \tag{14}$$

The continuous-time gradient descent dynamics for minimizing the square loss corresponds to the following ordinary differential equation (ODE):

$$\dot{\boldsymbol{\theta}}_t = -\nabla_{\boldsymbol{\theta}_t} \left( \frac{1}{2} \sum_{i=1}^{N} (y_i - f_{\boldsymbol{\theta}_t}(\mathbf{x}_i))^2 \right)$$

$$= \sum_{i=1}^{N} (y_i - f_{\boldsymbol{\theta}_t}(\mathbf{x}_i)) \nabla_{\boldsymbol{\theta}_t} f_{\boldsymbol{\theta}_t}(\mathbf{x}_i).$$

#### A.1.2. NEURAL TANGENT KERNEL

When analyzing the training dynamics of neural networks through the lens of kernel regression, a particular kernel known as the Neural Tangent Kernel (NTK) naturally arises. For a neural network $f_{\boldsymbol{\theta}}$ with $L$ hidden layers, the NTK is defined as:

$$K(\mathbf{x}, \mathbf{x}') = \left\langle \frac{\partial f_{\boldsymbol{\theta}}(\mathbf{x})}{\partial \boldsymbol{\theta}}, \frac{\partial f_{\boldsymbol{\theta}}(\mathbf{x}')}{\partial \boldsymbol{\theta}} \right\rangle. \tag{15}$$

This kernel function characterizes how the network output at two different input points changes with respect to the parameters during training. Under the gradient flow dynamics, we have:

$$\dot{f}_{\boldsymbol{\theta}_t}(\mathbf{x}) = \dot{\boldsymbol{\theta}}_t^\top \nabla_{\boldsymbol{\theta}_t} f_{\boldsymbol{\theta}_t}(\mathbf{x})$$

$$= \sum_{i=1}^{N} (y_i - f_{\boldsymbol{\theta}_t}(\mathbf{x}_i)) \nabla_{\boldsymbol{\theta}_t}^\top f_{\boldsymbol{\theta}_t}(\mathbf{x}_i) \nabla_{\boldsymbol{\theta}_t} f_{\boldsymbol{\theta}_t}(\mathbf{x}) \tag{16}$$

$$= K_t(\mathbf{x}, \mathbf{X})(\mathbf{Y} - f_{\boldsymbol{\theta}_t}(\mathbf{X})).$$

#### A.1.3. PROPERTIES OF NTK

For standard feedforward neural networks, the NTK can be computed recursively using:

$$K^{(\ell+1)}(\mathbf{x}, \mathbf{x}') = \sigma_w^2 \dot{\Sigma}^{(\ell)}(\mathbf{x}, \mathbf{x}') K^{(\ell)}(\mathbf{x}, \mathbf{x}') + \sigma_b^2, \tag{17}$$

where $K^{(\ell)}$ denotes the NTK at layer $\ell$, $\dot{\Sigma}^{(\ell)}$ represents the expectation of activation function derivatives, and $\sigma_w^2, \sigma_b^2$ are the variances of weights and biases, respectively.

In the infinite-width limit, the NTK exhibits several remarkable properties:

1. Pre-activations converge to a Gaussian process:

$$h^{(\ell)}(\mathbf{x}) \to \mathcal{GP}(0, K^{(\ell)}). \tag{18}$$

2. The empirical kernel converges to a deterministic kernel:

$$\lim_{n_1,\dots,n_L \to \infty} K_t(\mathbf{x}, \mathbf{x}') = K_0(\mathbf{x}, \mathbf{x}'). \tag{19}$$

3. The convergence rate is $\mathcal{O}(1/\sqrt{\min_\ell n_\ell})$ (Jacot et al., 2018).

### A.1.4. TRAINING DYNAMICS THROUGH NTK

The evolution of network outputs during training can be described through the NTK:

$$\frac{d}{dt} f_{\boldsymbol{\theta}_t}(\mathbf{x}) = -K_t(\mathbf{x}, \mathbf{X}) K_t^\dagger(\mathbf{X}, \mathbf{X})(f_{\boldsymbol{\theta}_t}(\mathbf{X}) - \mathbf{Y}). \tag{20}$$

As $t \to \infty$, the solution converges to:

$$f_{\boldsymbol{\theta}_\infty}(\mathbf{x}) = K_0(\mathbf{x}, \mathbf{X}) K_0^\dagger(\mathbf{X}, \mathbf{X}) \mathbf{Y}. \tag{21}$$

This result demonstrates that in the infinite-width limit, neural network training becomes equivalent to kernel regression with the NTK, providing a theoretical foundation for understanding the learning dynamics of deep neural networks.

### A.2. Dot-Product Property

For normalized inputs ($\|\mathbf{x}\|_2 = 1$), the NTK becomes a function of only the dot product between inputs:

$$K_0(\mathbf{x}, \mathbf{x}') = K_0(c), \quad c = \mathbf{x}^\top \mathbf{x}'. \tag{22}$$

**Lemma A.1.** *For any two unit vectors* $\mathbf{x}, \mathbf{x}'$ *on the unit sphere, considering a neural network with ReLU activation function, their pre-activations at each layer* $\ell$ *in an infinitely wide network follow a Gaussian process with covariance* $K^{(\ell)}(c)$ *depending only on* $c = \mathbf{x}^\top \mathbf{x}'$.

*Proof.* We prove this result for the case of ReLU activation function. We proceed by induction over layers under the joint infinite-width limit ($n_1, \dots, n_L \to \infty$).

**Base Case ($\ell = 1$):** The first layer's pre-activations:

$$h^{(1)}(\mathbf{x}) = \mathbf{W}^{(1)}\mathbf{x} + \mathbf{b}^{(1)} \tag{23}$$

with $\mathbf{W}_{ij}^{(1)} \sim \mathcal{N}(0, \sigma_w^2/d)$, $\mathbf{b}_i^{(1)} \sim \mathcal{N}(0, \sigma_b^2)$, yield covariance:

$$K^{(1)}(\mathbf{x}, \mathbf{x}') = \sigma_w^2 \mathbf{x}^\top \mathbf{x}' + \sigma_b^2 = \sigma_w^2 c + \sigma_b^2. \tag{24}$$

**Inductive Step ($\ell \to \ell + 1$):** Assume $h^{(\ell)}(\mathbf{x}) \sim \mathcal{GP}(0, K^{(\ell)}(c))$. For ReLU activation $\phi$, the post-activation covariance becomes:

$$\mathbb{E}[\phi(h^{(\ell)}(\mathbf{x}))\phi(h^{(\ell)}(\mathbf{x}'))] = \frac{K^{(\ell)}(c)}{2\pi}\left[\sqrt{1 - \left(\frac{K^{(\ell)}(c)}{K^{(\ell)}(1)}\right)^2} + \arcsin\left(\frac{K^{(\ell)}(c)}{K^{(\ell)}(1)}\right)\right], \tag{25}$$

where $K^{(\ell)}(1) = \sigma_w^2 + \sigma_b^2$ by normalization. This defines a new dot-product kernel $g^{(\ell)}(c)$.

The $(\ell + 1)$-th layer covariance:

$$K^{(\ell+1)}(c) = \sigma_w^2 g^{(\ell)}(c) + \sigma_b^2, \tag{26}$$

thus preserving the dot-product dependence.

**NTK Recursion:** Following (Jacot et al., 2018), the NTK decomposes as:

$$K_0(c) = \sum_{\ell=1}^{L} \left( \prod_{k=\ell+1}^{L} \dot{\Sigma}^{(k)}(c) \right) \Sigma^{(\ell)}(c), \tag{27}$$

where $\Sigma^{(\ell)}(c) = K^{(\ell)}(c)$ and $\dot{\Sigma}^{(\ell)}(c) = \mathbb{E}[\phi'(h^{(\ell-1)}(\mathbf{x}))\phi'(h^{(\ell-1)}(\mathbf{x}'))]$. In the infinite-width limit, both terms become deterministic functions of $c$ (Arora et al., 2019). Note that while this proof focuses on ReLU activation, similar results may hold for certain other activation functions under appropriate conditions. $\qquad \square$

**Theorem A.2.** *For any dot-product kernel $K(\mathbf{x}, \mathbf{x}') = K(\mathbf{x}^\top \mathbf{x}')$ defined on the unit sphere $\mathbb{S}^{d-1}$, if $K$ is continuous and positive definite, it can be decomposed into a power series with positive coefficients and their sum is finite:*

$$K(c) = \sum_{n=0}^{\infty} a_n c^n, \quad a_n > 0, \quad \sum_{n=0}^{\infty} a_n < \infty.$$

*Proof.* We prove this result in three steps:

**Step 1: Power Series Expansion**

For a continuous positive definite kernel on $[-1, 1]$, by Mercer's theorem, it has a uniformly convergent eigenfunction expansion. For dot-product kernels on the sphere, these eigenfunctions are polynomials, which gives us the power series expansion:

$$K(c) = \sum_{n=0}^{\infty} a_n c^n.$$

**Step 2: Positivity of Coefficients**

To prove $a_n > 0$, we use the positive definiteness of the kernel. For any $f \in L^2(\mathbb{S}^{d-1})$:

$$\int_{\mathbb{S}^{d-1}} \int_{\mathbb{S}^{d-1}} K(\mathbf{x}^\top \mathbf{y}) f(\mathbf{x}) f(\mathbf{y}) d\mathbf{x} d\mathbf{y} > 0.$$

Substituting the power series expansion and using the fact that $\|\mathbf{x}\|_2 = \|\mathbf{y}\|_2 = 1$:

$$\sum_{n=0}^{\infty} a_n \int_{\mathbb{S}^{d-1}} \int_{\mathbb{S}^{d-1}} (\mathbf{x}^\top \mathbf{y})^n f(\mathbf{x}) f(\mathbf{y}) d\mathbf{x} d\mathbf{y} > 0.$$

Since this inequality holds for all $f$ and the integrals are non-negative by construction, we must have $a_n > 0$ for all $n \geq 0$.

**Step 3: Convergence of Coefficients**

Finally, we prove that $\sum_{n=0}^{\infty} a_n < \infty$. Since $K$ is continuous on the compact set $\mathbb{S}^{d-1}$, it is bounded. In particular, $K(1)$ is finite. At $c = 1$, we have:

$$K(1) = \sum_{n=0}^{\infty} a_n.$$

Since all $a_n > 0$ and their sum equals the finite value $K(1)$, we conclude that $\sum_{n=0}^{\infty} a_n < \infty$. $\qquad \square$

This result guarantees that any dot-product kernel can be approximated arbitrarily well by a finite polynomial with positive coefficients.

# B. Experimental Details

### B.1. Ablation Studies

To better understand the impact of different components in PEAK, we conducted comprehensive ablation studies on three key aspects. All experiments in this section were conducted on the same task setting as in Section 4 - image reconstruction on the Jetplane image with combined random and structured missing patterns.

**Effect of Regularization Coefficient**   We investigated the impact of regularization coefficient $\lambda$ on model performance. We examined $\lambda \in \{10^{-5}, 10^{-4}, 10^{-3}, 10^{-2}, 10^{-1}, 1, 10, 10^2\}$. As shown in Figure 9(a), PEAK achieves optimal performance (PSNR around 23.5dB) when $\lambda$ is between $10^{-2}$ and $10^{-1}$. Both too small ($< 10^{-3}$) and too large ($> 1$) values of $\lambda$ lead to significant performance degradation, with PSNR dropping to around 14dB at $\lambda = 10^2$.

**Output Dimension of Regularization Network**   We analyzed the effect of varying the output dimension $r$ of the regularization network. As shown in Figure 9(b), we explored dimensions $r \in \{10, 50, 100, 200, 300, 400, 500\}$. The results indicate that PSNR improves significantly (from 19.5dB to 23.8dB) as $r$ increases from 10 to 100. However, further increasing $r$ beyond 100 yields diminishing returns and even slight performance degradation, with PSNR decreasing to around 22dB at $r = 500$. This suggests that $r = 100$ achieves the optimal balance between performance and computational cost.

**Choice of Activation Function**   We compared the performance of various activation functions including ReLU, Sigmoid, Tanh, Softmax, HardTanh, ELU, Leaky ReLU, and others. Figure 9(c) shows that most activation functions achieve PSNR values between 23-24dB, demonstrating relatively stable performance. Softmax shows marginally better stability and slightly higher average PSNR across multiple trials. Sigmoid and Softmin demonstrate competitive performance with minimal difference from Softmax. ReLU and LeakyReLU show slightly lower performance, potentially due to the vanishing gradient problem.

Our ablation studies revealed several key findings:

- The choice of regularization coefficient $\lambda$ significantly impacts model performance, with optimal values between $10^{-2}$ and $10^{-1}$.

- An output dimension of $r = 100$ achieves the best performance, with larger dimensions showing diminishing returns.

- Most modern activation functions perform well, with Softmax showing marginally better stability, followed closely by Sigmoid and Softmin.

These results provide insights into PEAK's optimal configuration and demonstrate our method's robustness across different parameter settings.

## B.2. Integration with Net-GD and Net-ADM

For both the Net-GD and Net-ADM frameworks, we replace the original deep decoder network with our INR as follows, where $m$ denotes the number of measurements and $n$ represents the length of the signal to be reconstructed.

**Net-GD Integration:**

1. The INR $f_{\boldsymbol{\theta}}$ directly maps coordinates to pixel values

2. Gradient descent updates follow:

$$\boldsymbol{\theta}_{t+1} = \boldsymbol{\theta}_t - \eta \nabla_{\boldsymbol{\theta}_t} \||\mathcal{A} f_{\boldsymbol{\theta}_t}(\mathbf{X})| - \mathbf{y}\|_2^2$$

   where $\mathcal{A} : \mathbb{R}^n \to \mathbb{R}^m$ is the Gaussian random matrix.

**Net-ADM Integration:**

1. The INR output is treated as the primal variable

2. The ADMM updates follow:

$$\mathbf{z}_{t+1} = \arg\min_{\mathbf{z}} \frac{\rho}{2} \|\mathcal{P}\mathbf{z} - f_{\boldsymbol{\theta}_t}(\mathbf{X}) + \mathbf{u}_t\|_2^2$$

$$\boldsymbol{\theta}_{t+1} = \arg\min_{\boldsymbol{\theta}} \frac{1}{2m} \left\| \sqrt{|\mathcal{A} f_{\boldsymbol{\theta}}(\mathbf{X})|^2 + \epsilon \mathbf{1}} - \sqrt{\mathbf{y}^2 + \epsilon \mathbf{1}} \right\|_2^2 + \frac{\rho}{2} \|\mathcal{P}\mathbf{z}_{t+1} - f_{\boldsymbol{\theta}}(\mathbf{X}) + \mathbf{u}_t\|_2^2$$

$$\mathbf{u}_{t+1} = \mathbf{u}_t + \mathcal{P}\mathbf{z}_{t+1} - f_{\boldsymbol{\theta}_{t+1}}(\mathbf{X})$$

where $\mathcal{A} : \mathbb{R}^m \to \mathbb{C}^m$ denotes the discrete Fourier transform operator and $\mathcal{P} : \mathbb{R}^n \to \mathbb{R}^m$ is the zero padding operator that extends signals to match target lengths. The Lagrangian multiplier is expressed as $\mathbf{u}$, while $\rho$ serves as the penalty parameter in the augmented Lagrangian framework. The data fidelity term $\frac{1}{2m} \left\| \sqrt{|\mathcal{A}f_{\boldsymbol{\theta}}(\mathbf{X})|^2 + \epsilon\mathbf{1}} - \sqrt{\mathbf{y}^2 + \epsilon\mathbf{1}} \right\|_2^2$ is a smoothing version of $\frac{1}{2m}\||\mathcal{A}f_{\boldsymbol{\theta}}(\mathbf{X})| - \mathbf{y}\|_2^2$, where $\mathbf{1}$ is a vector of all ones. We empirically used $\rho = 1$ and $\epsilon = 0.001$, and took $\mathbf{z}_T \in \mathbb{R}^n$ as the final reconstruction result.

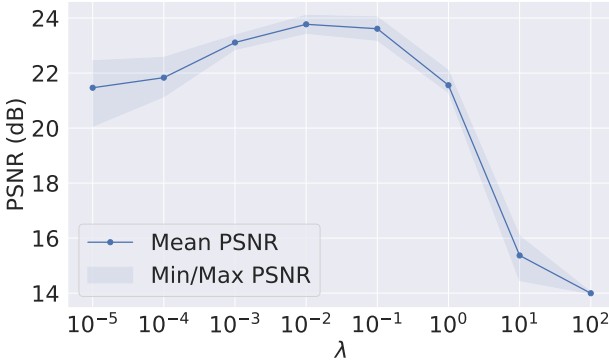

(a) Impact of regularization coefficient $\lambda$ on model performance. PEAK achieves optimal performance when $\lambda$ is between $10^{-2}$ and $10^{-1}$.

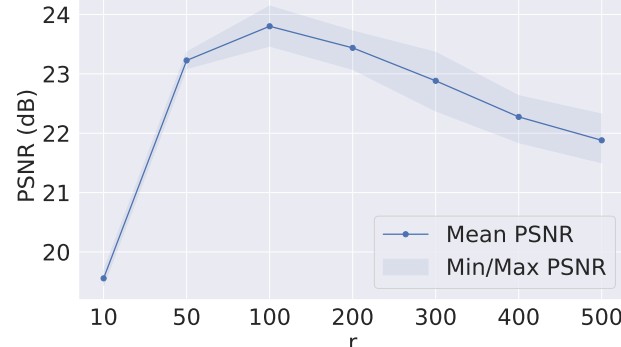

(b) Performance vs. output dimension of the regularization network. Output dimension of $r = 100$ achieves the best trade-off between performance and efficiency. The PSNR of the baseline on the Jetplane image is 22.56 dB. In most cases, the utilization of diverse output dimensions within the PEAK framework significantly surpasses this baseline performance.

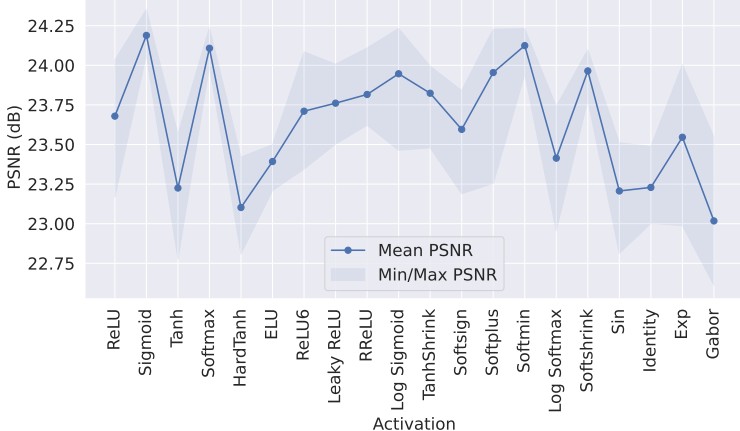

(c) Comparison of different activation functions. Softmax shows marginally better stability, followed closely by Sigmoid and Softmin.

*Figure 9.* Ablation studies on three key aspects of PEAK: (a) regularization coefficient $\lambda$, (b) output dimension of regularization network, and (c) choice of activation function. The results demonstrate the robustness of our method across different parameter settings and architectural choices.

