# OpenReview forum: "Learning Input Encodings for Kernel-Optimal Implicit Neural Representations"
_ICML.cc/2025/Conference — ICML 2025 poster_

### Official Review · Reviewer_i5ua · 2025-03-09

**Overall Recommendation:** 3

**Summary:**

This paper first established a theoretical insight that the neural tangent kernel of implicit neural representation can approximate any positive semidefinite dot-product kernels. Developing on this insight, the paper propose a kernel alignment regularizer to improve the INR system. Experiments show the proposed method performs better than baseline methods in image reconstruction and phase retrieval tasks.

**Claims And Evidence:**

110-114: The neural tangent kernel associated with an INR (which is a multi-layer perceptron) captures the evolution of network predictions during training. So by regularizing the NTK, we can control the training of INR. With this insight, the paper introduces a regularizer Eqn. (11) and claims this regularizer can improve the INR.

The evidence is both theoretical and empirical. The paper shows the proposed regularized loss function yield better performance in image reconstruction and phase retrieval.

The experiments indeed support the claim made by the paper. However, I personally consider the experiments to be a little simple. The experiments are simple tasks and mainly synthetic. If the proposed method can be applied to other more challenging real-world problems, the paper will be much stronger.

**Essential References Not Discussed:**

No

**Experimental Designs Or Analyses:**

As mentioned in the previous sections, the considered tasks seem to be too simple.

**Methods And Evaluation Criteria:**

The method proposed in the paper is simple and effective, and the theoretical foundation behind the regularization loss function is strong. So I believe the motivation and justification of the proposed method is valid. The proposed method is evaluated by two tasks and shows good results. But as mentioned in previous sections, the tasks seem to be too simple and work easily.

**Other Comments Or Suggestions:**

No

**Other Strengths And Weaknesses:**

No

**Questions For Authors:**

No

**Relation To Broader Scientific Literature:**

The proposed method seems to have a strong relation to reconstruction tasks, which are the foundation of many important tasks, such as unsupervised learning through reconstruction, image super resolution, etc.. From this perspective, the paper provides new insights into a fundamental problems, so I consider this as a significant contribution.

It would be much stronger if the proposed method is shown to be also effective in some real-world tasks, rather than only the synthetic tasks presented in this paper.

**Theoretical Claims:**

I didn’t carefully check the proof, while the overall idea makes sense to me.

---

> ### Author Rebuttal · Authors · 2025-04-01
>
> **All the Table/Figure Rx can be found in https://anonymous.4open.science/r/ICML-Re-3333**
>
> **Q1:**  The proposed method should be applied to other more challenging real-world problems.
> **A1:** Thank you for your valuable suggestion. We have conducted additional experiments on  more complex problems, specifically focusing on 3D neural fields and comparing our methods with Instant-NGP, as NeRF can be time-consuming. We performed experiments on the NeRF Synthetic datasets with 25, 50, and 100 view perspective samples. As shown in [Table R3](https://anonymous.4open.science/r/ICML-Re-3333/Table_R3.pdf), the average PSNR for 25 view perspectives indicates that the PEAK algorithm outperforms the vanilla Instant-NGP. Notably, the improvement is more pronounced with fewer training samples, demonstrating that PEAK effectively enhances generalization capability. Furthermore, [Figure R3](https://anonymous.4open.science/r/ICML-Re-3333/Figure_R3.pdf) shows that PEAK reduces artifacts caused by sparser samples, indicating its ability to leverage the internal structure of the data to improve performance in downstream tasks with limited data. We will include this 3D experimental data in the revised manuscript to further demonstrate the effectiveness and generalizability of our method.

---

### Official Review · Reviewer_DUFF · 2025-03-13

**Overall Recommendation:** 4

**Summary:**

The paper proposes two theoretically-motivated changes to implicit neural representation architecture and training, based on comparisons to the infinite width neural tangent kernel. The first change is a regularization strategy to encourage alignment with the optimal NTK, and the second is a trainable encoding that can be added before an INR to improve kernel alignment. Since the optimal kernel is not computable (due to both lack of access to the true data distribution and computational limitations of evaluating and inverting a large matrix), the idea is to encourage the INR to approximate this optimal kernel without computing it directly. Note that the proposed changes can be applied to any INR as a plug-in modification.

**Claims And Evidence:**

The claims of improved performance are well substantiated by experiments. The claims of matching the optimal kernel are substantiated with a toy experiment in Figure 2, though in general it seems the proposed method approximates a class of kernels that includes the optimal kernel, but may not exactly match the optimal kernel since that is in general not identifiable from a finite dataset.

**Essential References Not Discussed:**

N/A

**Experimental Designs Or Analyses:**

Please refer to my comments on methods and evaluation criteria. Specifically, I’d like to see some comparison or discussion of relative model sizes and training/evaluation times in addition to the provided comparisons of quality. I encourage the authors to release their code (upon publication, if not before).

**Methods And Evaluation Criteria:**

Evaluation on image inpainting (with different types of linear image corruption) and phase retrieval (a classic nonlinear inverse problem, also here for images), is quite compelling. These are reasonable examples to test the method and substantial improvement is shown compared to existing INRs, both quantitatively and qualitatively.

Although the quality of results obtained with the proposed method PEAK are impressive, it would be informative to also compare model size (ideally keeping the number of trainable parameters fixed across all models compared) and training/inference time for each method compared. Since PEAK requires adding a trainable embedding layer, I am concerned that it may gain an unfair advantage by increasing model size and/or training time.

**Other Comments Or Suggestions:**

Figure 1 is difficult to understand. Figure 1(a) could be improved by plotting with alpha<1 so that overlapping dots/lines can be distinguished. Figure 1(b) is not clear what it is trying to show (what do arrows mean? What are the level sets showing? What is the black triangle? etc.).

Overall the writing is clear, though there are occasional typos/minor grammatical mistakes. E.g. the sentence on line 214 (left column) is not a complete sentence, though the meaning is still clear.

**Other Strengths And Weaknesses:**

N/A; see the main review.

**Questions For Authors:**

My primary question for the authors is about the model sizes and training/inference times for each of the methods compared, to determine whether the improved performance comes at a cost. My leaning towards acceptance assumes that this information will be shared during the rebuttal/revision.

I also have some questions about notation in the theoretical contributions; these can be addressed by revising the exposition there.

**Relation To Broader Scientific Literature:**

Related work is adequately discussed.

**Theoretical Claims:**

The theoretical claims include a characterization of some properties that must be satisfied by the optimal kernel method for a dataset, and algorithmic and approximation choices to encourage an INR to approximately satisfy these same properties (including a theorem that such an INR exists). Since the optimal kernel itself is not identifiable from a discrete dataset, I would encourage the authors to relax some of the language around alignment with the optimal kernel since it seems the proposed method is instead endowing the INR with some properties that are also shared by the optimal kernel. Nonetheless, the theoretical contribution is valuable as most INR architectures make no attempt at theoretical motivation or characterization. Figure 2 does also validate in a toy setting that this approximation can induce an INR to mimic the optimal kernel in a setting where the optimal kernel is known and computable, which is a compelling illustration of the main idea.
Some notation is not clear. Specifically:
- What is A? In Theorem 3.3 A would seem to be a subset of the real numbers reflecting the range space of a kernel. Then A appears in section 3.4 where it would appear to be a function of two arguments, sometimes bold and sometimes not bold. From context my guess is that the bold version is a vectorized version of the non-bold version of A, but that neither A in section 3.4 is related to the A in theorem 3.3. Eventually the bold function A is defined in line 266 (right column), but this is after the reader has been seeing it without definition for almost a page. It would be preferable to explain at first use that A is a function to be learned, that must satisfy certain properties.
- The embedding function gamma is sometimes denoted with the internal parameters a_j listed explicitly, and sometimes with these parameters implicit in the function. This is a minor issue but at first glance can make it difficult to see that these are all the same function. Since gamma is also introduced and described before the practical choice for its definition/parameterization is given (in line 257, right column), I would suggest mentioning at first use that gamma is, like A, a function to be learned that must satisfy certain properties.
- The circled plus notation is used (e.g. on line 247, left column) without definition. The same notation can be used for multiple operations (e.g. direct sum, exclusive or, and dilation) so it warrants precise definition here.

---

> ### Author Rebuttal · Authors · 2025-04-01
>
> **All the Table/Figure Rx can be found in https://anonymous.4open.science/r/ICML-Re-3333**
>
> **Q1:** A comparison of model size and training/inference time, with a fixed number of trainable parameters, is necessary for a fair evaluation.
> **A1:** We have conducted a comparison regarding parameter numbers and training time. Our results demonstrate that PEAK converges quickly and achieves the highest PSNR at the same training time, as seen in [Figure R1](https://anonymous.4open.science/r/ICML-Re-3333/Figure_R1.pdf). Additionally, as shown in [Figure R2](https://anonymous.4open.science/r/ICML-Re-3333/Figure_R2.pdf), PEAK maintains the highest PSNR with an equivalent number of parameters compared to other methods. These findings indicate that our PEAK excels in both efficiency and performance.
>
> **Q2:** It may be beneficial to relax the language around alignment with the optimal kernel.
> **A2:** Thank you for your insightful feedback. We agree that complete alignment with the optimal kernel is unrealistic due to its unidentifiability from discrete data. Our method aims to endow the INR with properties shared by the optimal kernel, rather than asserting full alignment. In the revised paper, we will adjust our wording to more accurately reflect the actual capabilities of the method and avoid overemphasizing alignment.
>
> **Q3:** Some notation is not clear.
> (1) The notation for the variable A in Theorem 3.3 and Section 3.4 appears inconsistent and requires clarification.
> (2) The notation for the embedding function $\gamma$ requires clarification.
> (3) The notation for the circled plus is used without definition and can represent multiple operations.
> **A3:** Thank you for pointing out the confusion regarding the notations.
> (1) We sincerely apologize for the oversight. In fact, the $A$ in Theorem 3.3 and Section 3.4 represents entirely different concepts. In Theorem 3.3, $A$ denotes a subset of the real numbers, reflecting the range space of a kernel, whereas in Section 3.4, $\mathbf{A}:\mathcal{X}\times \mathcal{X}\rightarrow \mathbb{R}$ (denoted in bold) is a function that measures the similarity between input points. To enhance clarity, we will change the notation in Theorem 3.3 to signify a set, and we will move the definition of the function $\mathbf{A}$ in Section 3.4 to its first mention in the revised manuscript.
> (2) The initial definition of $\gamma$ is provided at line 166 in the right column. We will review the text to ensure clarity and reinforce the understanding that $\gamma$ is a learnable function with specific properties.
> (3) The $\oplus$ and $\otimes$ represent the direct sum and direct product, respectively. We will add more detailed explanations following these symbols.
>
> **Q4:** Encourage the release of the code upon publication, if not before.
> **A4:** We will make our code publicly available upon publication to facilitate reproducibility and further research in this area.
>
> **Q5:** The supplement on ablation studies and model architecture details should be included in the main paper.
> **A5:** We appreciate your suggestion regarding the supplementary material on ablation studies and model architecture details. We will ensure to reference this material in the main paper and summarize the key findings to provide readers with a clearer understanding of our experimental design.
>
> **Q6:** Improvements are needed for the clarity of Figure 1.
> **A6:** Thank you for your feedback on Figure 1. We will follow your suggestions to improve its clarity in the revised version. In Figure 1(a), we will reduce the transparency to better distinguish overlapping dots and lines. In Figure 1(b), the "orange arrows" indicate the training progress, i.e. training INR with the vanilla loss function and PEAK, resulting in $f_{\boldsymbol{\theta}}(\gamma(\cdot))$ and $f_{\hat{\boldsymbol{\theta}}}(\hat{\gamma}(\cdot))$, respectively. The "green arrows" represent the application of the NTK theorem to calculate the corresponding NTK values, where $f_{\boldsymbol{\theta}}(\gamma(\cdot))\rightarrow K$ (depicted as the black triangle) and $f_{\hat{\boldsymbol{\theta}}}(\hat{\gamma}(\cdot))\rightarrow \hat{K}$ (depicted as the orange star).  The "blue arrows" illustrate the theoretical analysis that introduces the optimal kernel $K^*$ (blue star). The vanilla INR's corresponding $K$ (black triangle) is far from the $K^*$ (blue star). Our KAR aligns the $\hat{K}$ with $K^*$ in the kernel space, guiding the INR to optimize $\gamma$. And the level sets represent the kernel space. We will add further explanation and simplify Figure 1(b) to enhance clarity in the revised version.
>
> **Q7:** Some typos/minor grammatical mistakes. E.g. the sentence on line 214 (left column) is not complete.
> **A7:** Thank you for pointing out the incomplete sentence. We will conduct a thorough review of the manuscript to correct these typographical and grammatical errors.

---

### Official Review · Reviewer_Z6av · 2025-03-14

**Overall Recommendation:** 3

**Summary:**

The paper summarises NTK-theory related contributions on INRs, and derives the optimal kernel for INRs under certain conditions. It then proposes an algorithm, named PEAK, to approximate a "Kernel Alignment Regularizer" and apply it to an INR, so that its kernel is encouraged to move towards the optimal one, improving its generalisation performance. Experiments show that the proposed approach achieves a similar kernel to the optimal one in a simple function approximation scenario. More practical experiments on image reconstruction and phase retrieval show that the proposed approach helps a simple ReLU-based MLP to achieve better results than with Fourier features or hash encoding.

##Update after rebuttal
The rebuttal addressed most of my concerns. I am inclined towards acceptance, and expect the authors to incorporate the new results/experiments/considerations in the final version of the paper in case of acceptance.

**Claims And Evidence:**

The theoretical claims and supported by proofs and analyses, however the experimental claims about the effectiveness of the proposed approach in improving the generalisation of INRs lack experimental support. See below.

**Essential References Not Discussed:**

See the experimental section of the review.

**Experimental Designs Or Analyses:**

Yes. While I find the experiments themselves to be appropriate for the evaluation of generalisation capabilities of the method, I believe that the shown results are not sufficiently convincing.
E1) SIREN [1] is mentioned in the paper but not used for comparisons in the experiments. Other more recent methods such as MFN[2], BACON [3], Gauss [4], WIRE [5], FINER [6] and SAPE [7] should be compared to, potentially also showing whether the proposed framework can be applied to such methods and improve them. Currently, the experiments convince the reader that the framework works on ReLU MLPs, but they do not support a practical use of the framework.
E2) On a similar note, time and memory requirements of the algorithm are not discussed, which would be needed to justify its practicality.

E3) Additional experiments on different tasks would also be appropriate, such as 3D shape and neural fields, the latter of which would greatly benefit from better generalisation capabilities.

E4) The supplementary shows some sensitivity to hyper parameter choice. This should be discussed in a limitations section in the main paper.

E5) Since INRs have been shown to have a bias towards low frequencies, how does the proposed regularizer affect this? An analysis would be interesting.

E6) Additionally, a second (compact) INR is used to compute the function A. How many additional parameters does this introduce? Does the proposed approach work better than a vanilla INR that has as many parameters as the total of f + g?

[1] Implicit Neural Representations with Periodic Activation Functions, Sitzmann et al.
[2] Multiplicative Filter Networks, Rizal Fathony, Anit Kumar Sahu, Devin Willmott, J Zico Kolter
[3] BACON: Band-limited Coordinate Networks for Multiscale Scene Representation, David B. Lindell, Dave Van Veen, Jeong Joon Park, Gordon Wetzstein
[4] Beyond Periodicity: Towards a Unifying Framework for Activations in Coordinate-MLPs, Sameera Ramasinghe, Simon Lucey
[5] WIRE: Wavelet Implicit Neural Representations, Saragadam et al.
[6] FINER: Flexible spectral-bias tuning in Implicit NEural Representation by Variable-periodic Activation Functions, Zhen Liu, Hao Zhu, Qi Zhang, Jingde Fu, Weibing Deng, Zhan Ma, Yanwen Guo, Xun Cao
[7] SAPE: Spatially-Adaptive Progressive Encoding for Neural Optimization, Hertz et al.

**Methods And Evaluation Criteria:**

The proposed method seems appropriate and well grounded.

**Other Comments Or Suggestions:**

Page 4 Line 212-215 column 1: sentence is not finished, the "while" could be removed.
Section 4.2: missing reference to figure 3, which is not referenced anywhere

**Other Strengths And Weaknesses:**

Strengths:
S1) The paper is overall well written
S2) The method seems original and potentially interesting to use in practice

Weaknesses:
W1) The experiments do not convince the reader about the validity of the method except with a very basic baseline. The number of additional parameters, as explained above, may also present a weakness as it is not currently discussed.
W2) Limitations are not discussed.

**Questions For Authors:**

I would like the authors to address my concern about experimental validity, in case I missed some points or scope.

**Relation To Broader Scientific Literature:**

The theoretical contributions seem well contextualised in the literature. However the experiments do not show relations to prior experimental work.

**Theoretical Claims:**

I checked the claims to the best of my abilities, however I could not verify the proofs due to my limited expertise in NTK theory

---

> ### Author Rebuttal · Authors · 2025-04-01
>
> **All the Table/Figure Rx can be found in https://anonymous.4open.science/r/ICML-Re-3333**
>
> **Q1:** The paper should compare PEAK with SIREN, MFN, BACON, Gauss, WIRE, FINER and SAPE, and explore whether the framework can be applied to and improve these methods.
> **A1:** Thank you for the concern regarding the experimental validity. Actually, we have shown that our PEAK algorithm is not sensitive to the choice of INRs with different activation functions, as illustrated in Appendix Figure 6(c). Furthermore, we have applied PEAK to the INRs you mentioned, except for SAPE, in *[Tables R1](https://anonymous.4open.science/r/ICML-Re-3333/Table_R1.pdf) and [R2](https://anonymous.4open.science/r/ICML-Re-3333/Table_R2.pdf)*. The exclusion of SAPE is due to its significantly different training strategy, which requires additional fine-tuning for a fair comparison. For more details about these experiments and our motivations, please refer to ***A1 to Reviewer QEM1***.
>
> **Q2:** The time and memory requirements of the algorithm are not discussed.
> **A2:** As shown in *Figure R1*, for the "Baboon" image reconstruction task, our proposed PEAK achieves a PSNR of 30 dB in 2 seconds, which benefits from the kernel alignment providing additional self-similarity to accelerate convergence. As shown in *Figure R2*, PEAK achieves a higher PSNR compared to other methods while using the same number of parameters. We will include a more comprehensive discussion of these results in the revised version.
>
> **Q3:** Additional experiments on tasks like 3D shape representation and neural fields.
> **A3:** Thank you for the suggestion. We have conducted more experiments on neural fields, comparing our methods with Instant-NGP, as NeRF can be time-consuming. Specifically, we performed experiments on the NeRF Synthetic datasets with 25, 50, and 100 view perspective samples, respectively. As shown in *Table R3*, the average PSNR for 25 view perspectives indicates that the PEAK algorithm outperforms the vanilla Instant-NGP. Notably, the improvement is more pronounced with fewer training samples, demonstrating that PEAK effectively enhances generalization capability. *Figure R3* shows that PEAK reduces artifacts caused by sparser samples, indicating its ability to leverage the internal structure of the data to improve performance in downstream tasks with limited data. We will include this 3D experimental data in the revised manuscript to further demonstrate the effectiveness and generalizability of our method.
>
> **Q4:** The supplementary shows some sensitivity to hyperparameter choice. This should be discussed in a limitations section.
> **A4:** Due to space constraints, we only discuss the polynomial degree of $\gamma$ in the main text, as it is unique to PEAK. The influence of other parameters (e.g., the regularization coefficient $\lambda$) is a common consideration for all regularization-based methods. The ablation study on activation functions demonstrates that PEAK is generally insensitive to these choices. While the output dimension $r$ of the regularization network appears to affect the final results, all configurations still outperform the baselines. We will provide the PSNR values of baselines in the supplementary material to avoid potential misunderstandings.
>
> **Q5:** An analysis of the proposed regularizer's impact on low-frequency bias in INRs is needed.
> **A5:** As shown in *Table R4*  and *Figure R4*, we conducted experiments using a synthetic image.  We sampled a $256\times 256$ grid from $[-1,1]\times [-1,1]$ as $\mathcal{X}$ and generated  $\mathcal{Y}$ using $\mathbf{y}_i=\sin(50\pi\sin(\frac{\pi}{3}\cdot\left\\|\mathbf{x}_i\right\\|_2))\in\mathcal{Y}$. We then trained MLP, Fourier, and our proposed PEAK algorithm on $\mathcal{X}\times\mathcal{Y}$. This synthetic image represents a frequency gradient transitioning from low (outer) to high (center, $(0,0)$). The results indicate that the MLP struggles to learn higher frequencies even after 10,000 epochs. The Fourier shows some improvement by initially capturing low frequencies before gradually learning high frequencies, but this process is relatively slow. In contrast, our PEAK algorithm effectively addresses the low-frequency bias, enabling it to learn both low and high frequencies almost simultaneously.
>
> **Q6:** Clarify the number of additional parameters introduced by the second (compact) INR and compare it to a vanilla INR with the same number of parameters.
> **A6:** The second (compact) INR introduces around additional $\frac{1}{10}$ of the first (main) INR parameters. We have compared its performance against a vanilla INR with the same total number of parameters. As illustrated in *Figure R2*, our algorithm exhibits superior performance.
>
> **Q7:** "While" should be removed; a reference to Figure 3 is missing.
> **A7:** Thank you for pointing out the incomplete sentence and the missing reference to Figure 3. We will make the necessary corrections in the revised manuscript.

---

### Official Review · Reviewer_QEM1 · 2025-03-14

**Overall Recommendation:** 2

**Summary:**

The paper studies Implicit Neural Representation (INRs) from a Neural Tangent Kernel (NTK) perspective. It introduces the *Kernel Alignment Regularizer* (KAR), which encourages alignment between the INR’s NTK and an optimal kernel and *Plug-in Encoding for Aligned Kernels* (PEAK).  PEAK is a method to integrate KAR with INR architectures with learnable input encodings. The authors show that PEAK improves image reconstruction and phase retrieval tasks compared to simple baselines (MLPs, Fourier features, and Hash encoding).

**Update after rebuttal.** Please refer to [my last comment](https://openreview.net/forum?id=Cx80t5FAQJ&noteId=zeP46QL4uf).

**Claims And Evidence:**

The paper makes convincing claims.

**Essential References Not Discussed:**

The paper is not the first work studying INRs from an NTK perspective. In particular, Yuce et al. (2022) should be discussed.

Yuce, Gizem, et al. *A structured dictionary perspective on implicit neural representations*. Proceedings of the IEEE/CVF Conference on Computer Vision and Pattern Recognition. 2022.

**Experimental Designs Or Analyses:**

The experiments show that PEAK improves over baselines, but it is unclear if the chosen baselines are strong enough. Moreover, the per-image results in tables in the main are unnecessary and could be aggregated.

**Methods And Evaluation Criteria:**

The methodology is reasonable, but some stronger baselines are lacking, e.g., SIREN, WIRE.

**Other Comments Or Suggestions:**

(L217) Notice that, for fully-connected networks, depth does not help generalization in the NTK regime (Bietti & Bach, 2020).

**Other Strengths And Weaknesses:**

The paper is dense, making it hard to follow. For instance, proofs could be moved to the appendix to improve readability, while providing in the main text only clearer high-level intuitions.

**Questions For Authors:**

1. What motivated the choice of the current baselines?

**Relation To Broader Scientific Literature:**

To the best of my knowledge, the proposed PEAK algorithm is novel. However, several theoretical results – presented as novel – are known or well-established in the literature. For instance, all results reported in Appendix A are not novel and not referenced. Moreover, the result in Theorem 3.1 expressing the optimal kernel in terms of the posterior mean kernel is a standard result in Gaussian process and Bayesian nonparametric methods. In particular (6) is the Bayesian optimal kernel which minimizes the MSE.

**Theoretical Claims:**

Proofs appear correct. Several theoretical results were already known and not properly referenced.

---

> ### Author Rebuttal · Authors · 2025-04-01
>
> **Q1:** The choice of the current baselines needs clarification.
> **A1:** We appreciate your inquiry regarding the choice of baselines. Our PEAK algorithm is designed to find an encoder $\gamma(\mathbf{x})$ that enhances the generalization ability of the composed function $f_{\boldsymbol{\theta}}(\gamma(\mathbf{x}))$. Consequently, we focus on these baselines that also work with $\gamma(\mathbf{x})$. The vanilla MLP with $\gamma(\mathbf{x})$ serves as a benchmark for basic performance.  The Fourier feature and Hash encoding are among the most influential methods for improving high-frequency representation capability and convergence speed. In response to your feedback, we have extended Table 1 to include a broader range of baselines in [Table R1](https://anonymous.4open.science/r/ICML-Re-3333/Table_R1.pdf), such as SIREN, MFN, BACON, Gauss, WIRE, and FINER. While these more recent methods show improved high-frequency representation ability in their original studies, their generalization ability remains limited, as shown in [Table R1](https://anonymous.4open.science/r/ICML-Re-3333/Table_R1.pdf). Additionally, our KAR can be directly applied to these baselines in a plug-and-play manner. We further evaluate the improvement of KAR on the aforementioned baselines in [Table R2](https://anonymous.4open.science/r/ICML-Re-3333/Table_R2.pdf). The results indicate that KAR enhances the performance of all these baselines, and the numerical results are consistent and not sensitive to the choice of baseline, as KAR estimates the same optimal kernel under the same sampling pattern and image.
>
> **Q2:** Some theoretical results were not properly referenced.
> **A2:** Thanks for your insightful suggestions from the perspective of a peer in this specialized field. Our main contributions are the introduction of KAR regularization and the PEAK algorithm, both derived from Theorem 3.3. Given the wide-ranging applications of INR across various research domains, our goal is to present the core ideas in a manner that is accessible to researchers with varying levels of prior knowledge, without necessitating consultation of the original literature. Therefore, we have distilled the essential part of these Theorems. In the revised manuscript, we will include proper citations in Appendix A to facilitate readers in finding the related works.
>
> **Q3:** The per-image results in tables in the main are unnecessary and could be aggregated.
> **A3:** Thank you for pointing this out. We believe that including more image results is important as the performance of INRs is closely tied to the specific characteristics of each image. For instance, in Table 1, PEAK shows the highest improvement on the "Baboon" image with a missing patch, due to its left-right symmetry. This symmetry allows PEAK to enhance generalization by effectively learning internal self-similarities within the signal. A more detailed discussion on this will be provided in the revised version.
>
> **Q4:** The relationship of the work to the findings of Yuce et al. (2022) should be discussed.
> **A4:** NTK is a useful mathematical tool for analyzing the dynamics of INRs, and many researchers have utilized it to illustrate the properties of INRs. Yuce et al. (2022) analyze INRs from a dictionary learning perspective, which differs significantly from our approach. In contrast, Tancik et al. (2020) provide an earlier analysis of INRs from the NTK perspective, proposing the Fourier feature encoder to improve high-frequency representation capabilities, which we have cited in our paper. To our knowledge, our PEAK is the first work to employ kernel alignment for guiding encoder design. In the revised manuscript, we will include a citation to clarify the similarities and differences between our work and that of Yuce et al. (2022).
>
> **Q5:** The readability of the paper could be improved.
> **A5:** We recognize that the paper's density may pose a challenge to readers. To enhance readability, we will move technical proofs to the appendix and focus on providing clearer high-level intuitions in the main text, such as summarizing key concepts and illustrating them with examples. Additionally, we will work on simplifying the exposition of our methods and results.
>
> **Q6:** Notice that, for fully connected networks, depth does not help generalization in the NTK regime (Bietti & Bach, 2020).
> **A6:** Thank you for your valuable comment regarding the findings of Bietti & Bach (2020). We agree that while depth may not theoretically improve generalization in the NTK regime for fully connected networks, in practical applications—especially when the network width is not infinite—depth can still positively influence model performance. In fact, there is very limited research that relies solely on a single-layer INR for realistic applications. Therefore, we propose an encoder learning algorithm rather than modifying the activation function in a single-hidden-layer INR, as done by Simon et al. (2022).

---

> > ### Comment · Reviewer_QEM1 · 2025-04-08
> >
> > I thank the authors for their answers and the additional results provided. However, since the updated manuscript cannot be reviewed, I believe the paper requires another round of review to properly assess (i) whether all prior theoretical results are properly referenced and clearly presented as such, and (ii) the new baselines and experimental findings.

---

> > > ### Author Response · Authors · 2025-04-09
> > >
> > > Thank you for your thorough review of our manuscript. We would like to clarify our position regarding the request for another round of review:
> > >
> > > (i) **Referencing Theoretical Results**: The NTK theory referenced in the main text (L122, left side) has been appropriately cited, and we have not intentionally overlooked relevant works to obscure our theoretical contributions. Additionally, we have cited one of the most representative works, Tancik et al. (2020), in multiple places (e.g., L46, right side; L166, right side) and have thoroughly discussed its relationship to our work. Specifically, we have compared the performance of the fixed input Fourier mapping (Fourier) proposed based on theoretical analysis with our learnable mapping in Tables 1, 2 and Figures 4, 5.
> > >
> > > (ii) **New Baselines and Experimental Findings**: The experiments we provided in the rebuttal are extensions of the original experiments. Most of the "new baselines" you mentioned involve modifications to the activation functions used in INRs. In fact, our original manuscript already included a comparison of up to 20 different activation functions in Appendix Figure 6(c), which encompasses the mentioned new baselines, including SIREN (corresponding to the sine activation function) and WIRE (corresponding to the Gabor activation function). The experimental results demonstrate that our algorithm is not sensitive to the specific network architecture. This experimental finding was already presented in the original manuscript, and the additional results in Tables R1 and R2 further reinforce this finding rather than indicating a new one.
> > >
> > > In summary, the theoretical discussions and experimental results provided in the rebuttal are consistent with those in the original manuscript, serving to enhance the robustness of our arguments. Based on these points, we believe that:
> > > 1. The original manuscript has adequately cited the most relevant works (e.g., Tancik et al. 2020, Jacot et al. 2018).
> > > 2. The additional experiments conducted were in response to the reviewer's request to further clarify the performance of our method, without introducing new baselines, altering conclusions, or presenting novel findings.
> > >
> > > We kindly request a fair evaluation of our original manuscript in light of the points we have clarified.

---

### Decision · Program_Chairs · 2025-05-01

**Decision:**

Accept (poster)

**Comment:**

This paper develops a theoretically grounded approach to improving implicit neural representations (INRs) via kernel alignment considerations, combining a Kernel Alignment Regularizer (KAR) and a learnable encoding strategy (PEAK). The reviewers agree that the core idea—harnessing Neural Tangent Kernel (NTK) analysis to guide design choices for INRs—is both original and well-motivated. They also commend the thorough theoretical underpinnings, alongside clear empirical gains on image restoration and phase retrieval tasks. Early concerns included the need for broader baseline comparisons, requests for more complex tasks beyond toy and 2D settings, and clarifications on whether the authors had properly cited related theoretical results. In their rebuttal, the authors provided additional experiments, compared against extended sets of recent methods, clarified that the theoretical results align with known NTK findings, and improved the manuscript’s explanations of key notation and motivations.

Overall, the reviewers now lean in favor of acceptance. While a few felt that the original draft could have used additional experiments and clearer presentation of theoretical references, the authors’ revisions and rebuttal materials have addressed most concerns. The consensus is that the paper makes a solid contribution to the literature, with strong theoretical grounding and evidence of practical benefits. The authors are encouraged to incorporate the newly provided experiments and clarifications into the final version for better readability and completeness.